# Imperceptible energy harvesting device and biomedical sensor based on ultraflexible ferroelectric transducers and organic diodes

Andreas Petritz [1,2], Esther Karner-Petritz[1,2], Takafumi Uemura [1,3], Philipp Schäffner [2], Teppei Araki[1,3], Barbara Stadlober [2✉] & Tsuyoshi Sekitani [1,3✉]

Energy autonomy and conformability are essential elements in the next generation of wearable and flexible electronics for healthcare, robotics and cyber-physical systems. This study presents ferroelectric polymer transducers and organic diodes for imperceptible sensing and energy harvesting systems, which are integrated on ultrathin (1-μm) substrates, thus imparting them with excellent flexibility. Simulations show that the sensitivity of ultraflexible ferroelectric polymer transducers is strongly enhanced by using an ultrathin substrate, which allows the mounting on 3D-shaped objects and the stacking in multiple layers. Indeed, ultraflexible ferroelectric polymer transducers have improved sensitivity to strain and pressure, fast response and excellent mechanical stability, thus forming imperceptible wireless e-health patches for precise pulse and blood pressure monitoring. For harvesting biomechanical energy, the transducers are combined with rectifiers based on ultraflexible organic diodes thus comprising an imperceptible, 2.5-μm thin, energy harvesting device with an excellent peak power density of 3 mW·cm$^{-3}$.

[1] The Institute of Scientific and Industrial Research, Osaka University, Ibaraki, Osaka, Japan. [2] JOANNEUM RESEARCH Forschungsgesellschaft mbH, MATERIALS-Institute for Surface Technologies and Photonics, Weiz, Austria. [3] AIST Advanced Photo-Bio Lab, Photonics Center Osaka University, Suita, Osaka, Japan. ✉email: barbara.stadlober@joanneum.at; sekitani@sanken.osaka-u.ac.jp

Many disruptive digital technologies like the Internet of Everything, cyber-physical systems, robotics or e-health are based on components that are inexpensive and facile to produce[1–3], made of sustainable and/or biocompatible materials[1], and are energy-saving, or even self-powered[1,4]. But only if constructed in a mechanically flexible[1,5], stretchable[5–11] or very thin[7,12] way, they allow for unobtrusive and seamless integration on machines, objects or the human body, often in the form of an electronic skin[4].

A perfect example is next-generation biomedical devices for accurate monitoring of physiological and vital parameters that can be conformably attached to human skin[7,13–18] or, in some cases, can even be implanted inside the body[19–21]. Since user compliance is greatly improved when such devices are comfortable to wear and not perceived as disturbing, bulky external power supplies and wiring should be avoided. Accordingly, next-generation biomedical devices must not only snuggle perfectly to the skin or tissue and be lightweight but should also be energy-autonomous. This can be provided by device-integrated nanogenerators and energy storage elements that will guarantee the continuous and imperceptible recording of medical parameters[4].

Although numerous examples of nanogenerators for wearable[4] and implantable[22] biomedical devices were presented, some of them also allowing for conformal contact to the skin[13,23,24], they lack of combining conformability, energy harvesting and sensing with a facile, scalable and cost-effective route to mass manufacturing[1]. A scalable method for the easy realisation of compliant sensors, nanogenerators and energy-storage elements is to integrate them on ultrathin substrates by spin coating or printing, thus making them ultraflexible.

Ultraflexible devices are devices with a total thickness below 10 μm[25] or a minimum bend radius <1 mm[26]. These characteristics allow them to be wrapped around or attached to moving and complex three-dimensional surfaces.

To date, few ultraflexible nanogenerators were realised for triboelectric[27] and solar energy conversion[25,28–31]. However, so far neither piezoelectric nanogenerators (PENGs) nor rectifying diodes have been reported that are truly ultraflexible according to the above definition.

Organic solar cells provide a higher power output as compared to tribo-[32] and piezoelectric nanogenerators[1,23], however, the latter have other favourable aspects[4]. In particular, the piezoelectric nanogenerator architecture is not just limited to energy generation but can simultaneously be exploited for multimodal sensing[1,4,23]. The ability of piezoelectric materials to respond to all types of mechanical stimuli provided by the human body, such as movement, pulse, air vibration or in/exhaling with high sensitivity, reproducibility, large dynamic range and short response time, makes them ideal candidates to be implemented in self-sustaining electronic medical devices for recording vital signals[4]. Furthermore, they can also be used as multimodal transducers in many other applications connected to IoT, robotics or cyber-physical systems[1,33,34]. A piezoelectric transducer material that can be effectively utilised in both PENGs and sensors is the ferroelectric co-polymer Poly(VinylideneDiFluoride:TriFluoroEthylene) P(VDF:TrFE)[1,33] processable from solution on flexible[35–39] or even stretchable substrates[40] by spin coating, printing or electrospinning[1,33].

A complete piezoelectric energy-harvesting device does not only consist of the nanogenerator but needs a rectifier circuit and an energy-storage element. Organic diodes represent good candidates for energy rectifiers, as they can be manufactured by thermal evaporation[41,42], batch printing[43,44] or even roll-to-roll printing techniques[45]. These organic diodes are created either using a vertical capacitor-like setup with an organic semiconductor sandwiched between two different electrodes[41,42], or in an organic thin-film transistor (OTFT) setup with shorted drain-gate electrodes[46–48]. However, the reported diodes do not have flexible or stretchable substrates[41,46,47] or they have substrates with only limited flexibility[43–45,48], which makes them inappropriate for use as rectifier circuits in conformable energy-harvesting systems.

We considered the abovementioned issues and demonstrated for the first time, a comprehensive and ultraflexible energy-harvesting device (UEHD) consisting of a piezoelectric nanogenerator, a diode-based rectifier and a storage capacitor; none of the components had a thickness larger than 2.5 μm. They were fabricated on a just 1-μm thin substrate made of parylene thus being imperceptible when attached to a surface e.g., to the human skin, which we demonstrate for a ferroelectric wireless e-health patch that can monitor pulse rate and measure the blood pressure (Fig. 1). Both the nanogenerators as well as the sensors are based on ultraflexible ferroelectric polymer transducers (UFPTs) from P(VDF:TrFE)$_{70:30}$ combined with ultraflexible full-wave rectifier circuits made from organic diodes in shorted OTFT geometry and ultraflexible thin-film capacitors using nanometre thin layers of alumina. The diodes, deploying DNTT (dinaphtho-thieno-thiophene) as the organic semiconductor, show very high-rectification ratios of over $10^7$ and due to their small transition voltage below 100 mV, a negligible built-in voltage drop. The UFPTs feature (i) ultraflexibility, which enables conformal attachment to various materials and surfaces and allows to stack multilayer UFPTs even on 3D-shaped carriers, (ii) high sensitivity to out-of-plane (transversal) stress that can be boosted by multilayer stacking thus achieving peak sensitivities in the range of 15 nC N$^{-1}$ for a three-layer stack on a pre-bent rubber carrier, (iii) excellent mechanical stability allowing for robust operation under a bending radius of 40 μm and (iv) very short response times far below 20 ms N$^{-1}$. When used as a PENG the UFPTs delivered peak power densities of over 3 mW cm$^{-3}$.

## Results

**Ultraflexible ferroelectric polymer transducer**. Ultraflexible ferroelectric transducers were fabricated on a 1-μm thin parylene diX-SR (Daisan Kasei Co., Ltd.) substrate. As shown in Figs. 1 and 2a, the capacitor-like transducer structure comprises a spin-coated ferroelectric co-polymer P(VDF:TrFE)$_{70:30}$ sandwiched in layers between two thermally evaporated metal electrodes. This ferroelectric polymer film is semi-crystalline with ferroelectric crystalline domains embedded in an amorphous matrix. Since polar domains are randomly oriented directly after spin coating, we need to apply an external electric field larger than the coercive field of the material to align dipole moments in the entire sample volume and thereby induce a macroscopic polarisation. This process is called 'electrical poling,' and it is described in detail in "Methods" and Supplementary Fig. 1. A representative 'poling curve' (the dependence of electric displacement $D$ on electric field $E$, $D(E)$) of a P(VDF:TrFE)$_{70:30}$ transducer is shown in Fig. 2b. This curve forms a hysteresis loop that reveals the bi-stable nature of the polarisation intrinsic to any ferroelectric. The two main figures-of-merit can be deduced from the ferroelectric $D(E)$-hysteresis: remnant polarisation $P_r$, defined as $P_r = |D(E = 0)|$, and coercive field $E_c$, defined as $E_c = |E(D = 0)|$, as indicated in Fig. 2b. Typical values for the UFPT figures-of-merit are $P_r = 65$ mC m$^{-2}$ and $E_c = 50$ V μm$^{-1}$. The macroscopic remnant polarisation of the ferroelectric co-polymer P(VDF:TrFE) is a measure of its sensitivity and the magnitude of the piezo- and pyroelectric coefficients[49]. The nearly rectangular shape of the hysteresis curve reveals a negligibly small difference between the $P_r$ and the saturation polarisation (i.e., the displacement at the maximum field). This indicates that the active ferroelectric layer has

## Ultraflexible Piezoelectric Energy Harvesting and Sensing Device

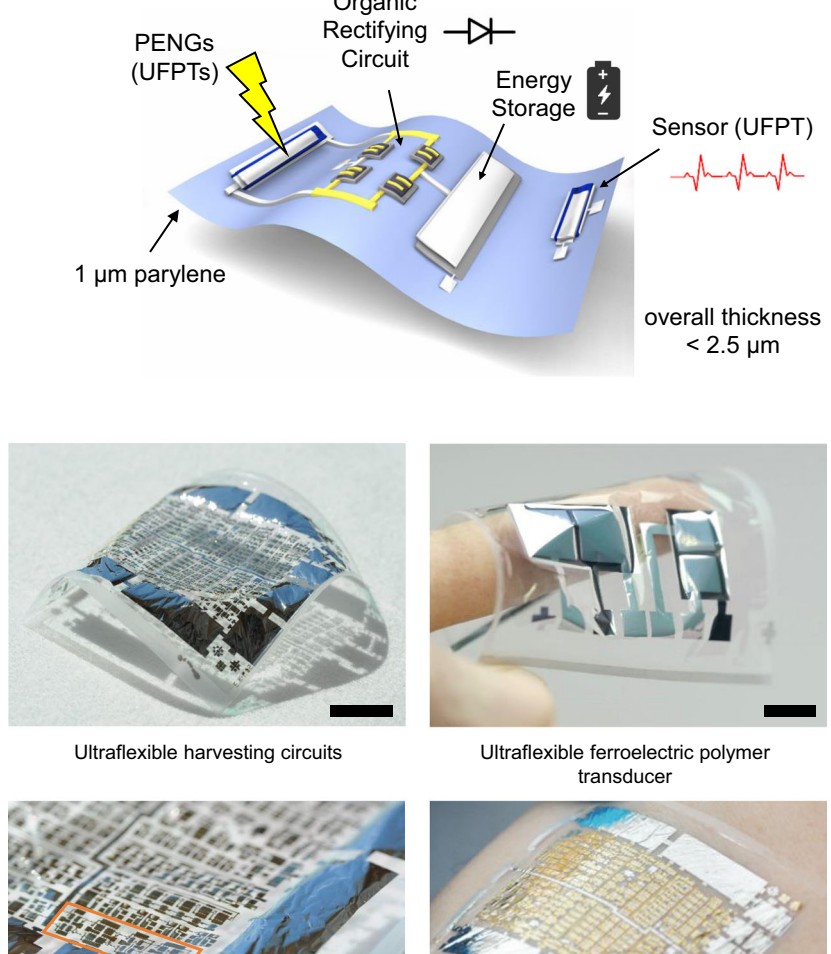

**Fig. 1 Ultraflexible piezoelectric energy harvesting and sensing.** Scheme of a range of ultraflexible devices integrated on a 1-µm thin parylene substrate and photographs of the fabricated devices. Ultraflexible ferroelectric polymer transducers (UFPTs) are used for vital parameter sensing and as piezoelectric nanogenerators (PENGs) for harvesting biomechanical energy when attached to the skin. For the comprehensive ultraflexible energy-harvesting device (UEHD), the ultraflexible nanogenerators are combined with ultraflexible circuits comprising organic diodes as rectifiers and thin-film capacitors for energy storage. Scale bar, 1 cm.

excellent dielectric properties (low dielectric losses) and that the crystalline dipoles display good switching behaviour.

The magnitude of $P_r$ is most strongly affected by the degree of crystallinity $X_c$ of the ferroelectric layer, as would be naturally expected because the spontaneous polarisation (and thus $P_r$) originates from the polar molecular crystallites only[50]. One way to enhance the crystallinity of the ferroelectric layer is to perform thermal annealing, which was reported to be most effective in the temperature region between the Curie temperature $T_C$ and the melting temperature $T_M = 153\,°C$ of P(VDF:TrFE)[1,50,51]. We tempered the ferroelectric layer directly after spin coating by placing the sample on a hot plate for 5 min and then placing it in a vacuum oven for 1 h (at the same temperature). Finally, we removed the sample from the oven and allowed it to cool to 30 °C

(RT) on a metal plate. As shown in Fig. 2c, $P_r$ reached a maximum of $P_{r,max}$ ~67 mC m$^{-2}$ at $T_A$ of 130 °C which is above $T_C$ (determined to be $\approx 105\,°C$ as deduced from the phase transition of the permittivity $\varepsilon_r$ in Supplementary Fig. 2a) and below $T_M$. More details of the ferroelectric-to-paraelectric phase transition and its relation to electric poling are shown in Supplementary Note 1 and in Supplementary Fig. 2.

The AFM images (topography and non-contact atomic force microscopy images; Supplementary Fig. 3) revealed that differences in the annealing temperature resulted in large differences in the surface morphology of the P(VDF:TrFE)$_{70:30}$ co-polymer thin films. Starting from a $T_A$ of ~ 90 °C, the size and density of the ferroelectric crystallites increased continuously until the temperature reached ~130 °C (inset in Fig. 2b, compare

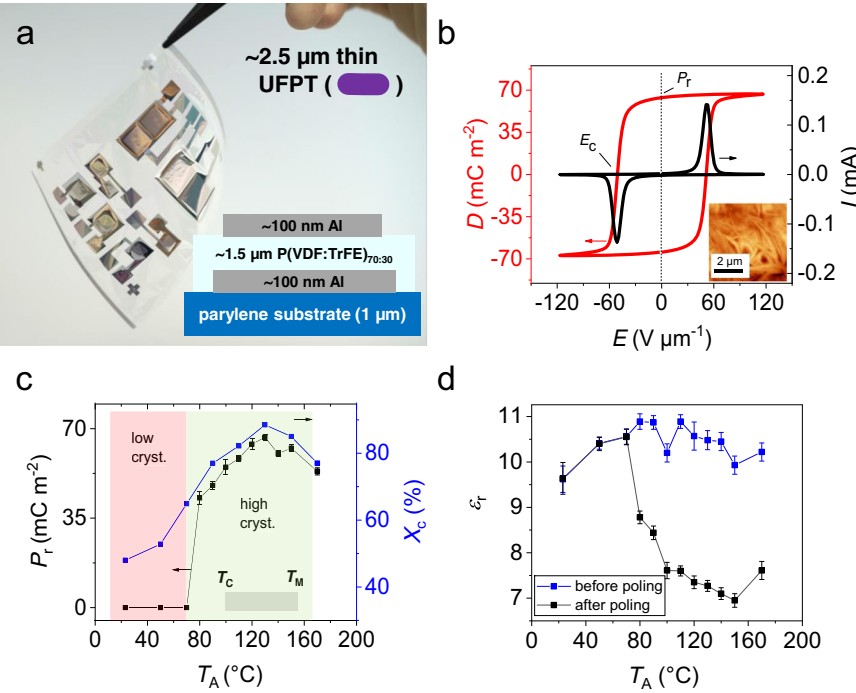

**Fig. 2 Ultraflexible ferroelectric polymer transducer (UFPT) setup, and the ferroelectric and dielectric properties. a** Photograph of the ultraflexible P(VDF:TrFE)$_{70:30}$-based transducer with a 1-μm thin diX-SR (parylene) substrate and the illustration of its setup. **b** Representative $D(E)$ hysteresis curve of the ferroelectric layer measured during poling at 1 Hz after annealing at 130 °C. $E_c$ denotes the coercive electric field strength where most microscopic dipoles start to rearrange themselves under the presence of an applied external field, and $P_r$ is the remnant polarisation in the absence of an external field (i.e., $D(E = 0 \text{ V μm}^{-1})$), which is the main figure-of-merit for the transducer. **c** $P_r$ and degree of crystallinity $X_c$ as a function of the annealing temperature $T_A$. The melting point ($T_M$) was reported earlier to be 153 °C[71] and the Curie temperature $T_C$ was measured to be ~ 105 °C (see Supplementary Fig. 2). **d** The dependence of $\varepsilon_r$ (mean value) on $T_A$ measured before and after poling. The displayed values of $P_r$ and $\varepsilon_r$ for each $T_A$ in (**c**) and (**d**) are mean values with standard deviations determined from at least ten devices with layer thickness values between 1.3 and 1.5 μm.

Supplementary Fig. 3). As obvious from Supplementary Fig. 3, the crystallites initially appeared as elongated grains, and then, as $T_A \geq 110$ °C, these transformed into the characteristic lamellar structure[52]. A high density of ferroelectric crystallites was observed in the AFM at $T_A = 130$ °C and $T_A = 150$ °C. Even after the layer had been annealed at about 170 °C, which is well above the melting temperature, lamellar structures were clearly observed after the sample had cooled to RT.

The dependence of $X_c$ on the annealing temperature was quantitatively investigated in more detail by taking XRD measurements (Fig. 2c, Supplementary Figs. 4 and 5a). The diffraction peak observed around $2\Theta = 20°$ is attributed to the (002)/(110)-plane reflections of the $\beta$-phase of P(VDF:TrFE), whereas the broad shoulder seen at lower angles is associated with the scattering from the molecules in the amorphous phase (Supplementary Fig. 5a)[51,53]. By fitting two Gaussian curves to the diffraction curve in the range $15° < 2\Theta < 25°$, as illustrated in Supplementary Fig. 5a, the crystallinity $X_c$ was derived as the ratio between the crystalline phase-related scattering intensity (derived from the (002)/(110) peak area), and the total scattering intensity from both the crystalline and amorphous phases. Supplementary Fig. 5a illustrates how the intensity was calculated from the area under the Gaussian curves, with the more detailed information provided in the Methods section. This approach provides an estimation of the crystallinity in a polymer and is referred to as 'apparent' crystallinity in literature[53].

The results of the XRD, AFM and $D(E)$ investigations allowed us to draw a clear correlation between the crystallinity and the remnant polarisation (Fig. 2c and Supplementary Fig. 5c). The maximum $X_c$ of $X_{c,max}$ ~84% was observed for $T_A = 130$ °C, which corresponds to the temperature at which the remnant

polarisation reached its maximum ($P_{r,max} = 67 \pm 2 \text{ mC m}^{-2}$, mean value and standard deviation calculated for ten transducers). At higher (140 °C $\leq T_A \leq$ 170 °C) or lower (80 °C $\leq T_A \leq$ 120 °C) annealing temperatures, the crystallinity was lower, showing the same trend as $P_r$. At annealing temperatures above 140 °C, the rectangular shape of the hysteresis loops was clearly distorted, and the saturation polarisation was significantly larger as compared to the remnant polarisation (Supplementary Fig. 4b). This indicates that the polarizability and, therefore, the mobility of the microscopic dipoles had increased, which is a result consistent with the slight observed increase in the relative permittivity $\varepsilon_r$ of the poled samples at temperatures above 150 °C (Fig. 2d). Below $T_A = 70$ °C, $X_c$ became so low that $P_r$ vanished. Thus, the threshold of the degree of crystallinity necessary for a ferroelectric behaviour seemed to occur around $X_{c,min}$ ~65%.

Moreover, impedance spectroscopy measurements indicated that $X_c$ is related to the dielectric properties of the ferroelectric co-polymer layer (Fig. 2d and Supplementary Fig. 5b). Supplementary Fig. 5b shows that the value of the capacitance of a poled sample was significantly lower than its value before poling, whereas its frequency dependence remained essentially unchanged. If we examine $\varepsilon_r$ (derived from impedance measurements) as a function of $T_A$ (Fig. 2d), we see that a poling-induced decrease in $\varepsilon_r$ occurred only above the threshold temperature of $T_A$ ~70 °C, where $X_c$—and consequently the number and size of crystallites—is drastically increased and ferroelectric behaviour is observed. Thus, we can conclude that the polar polymer chains of the crystallites that reorient themselves during electrical poling are less polarisable than the amorphous regions of the crystallites, which resulted in a reduced net permittivity. In other words, a strong decrease in $\varepsilon_r$ caused by electrical poling

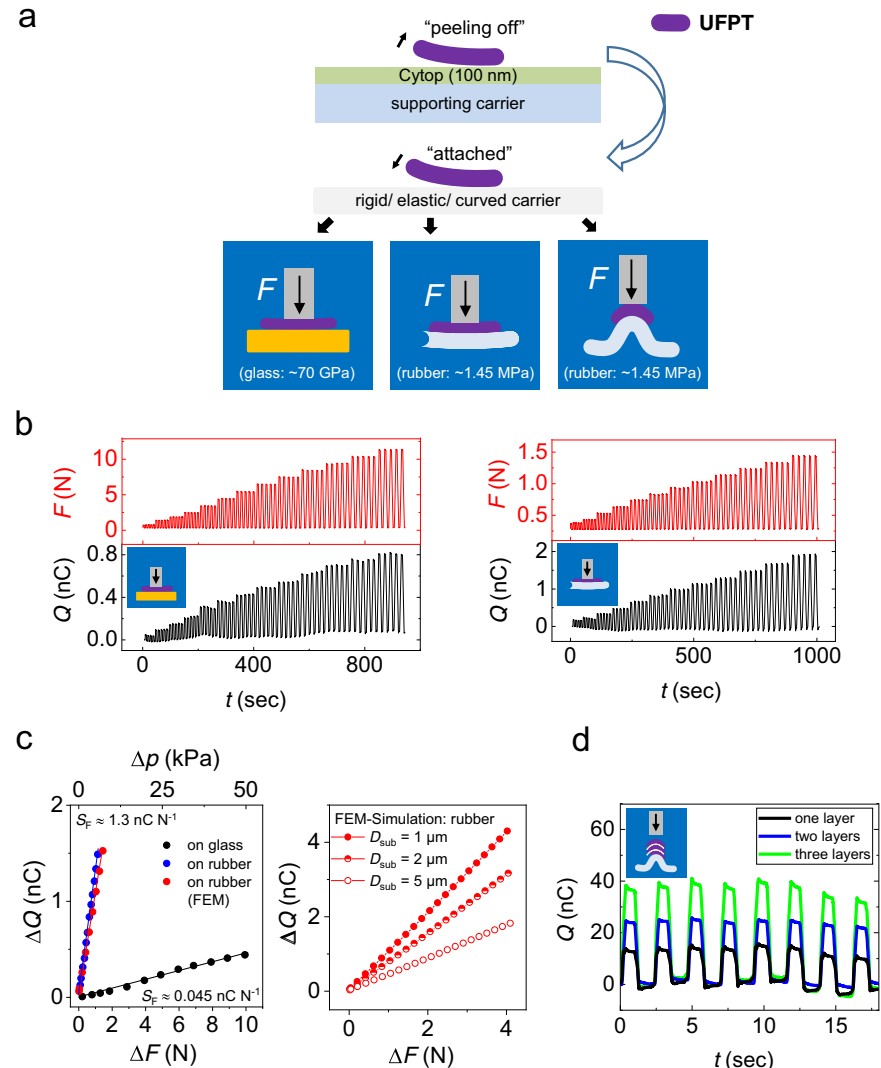

**Fig. 3 Transversal-load test on different carrier substrates. a** During fabrication, UFPT films are fixed on a supporting glass carrier. Later, the UFPT films can be easily peeled off the supporting glass carrier and applied to various carriers with different structure, shapes, curvatures or material mechanic properties. To perform transversal-load testing, the UFPTs were applied to one of two types of carriers: a 1-mm-thick rigid flat glass carrier (Young modulus $Y$ ~70 GPa) or a 6-mm-thick elastic silicone rubber carrier ($Y$ ~1.45 MPa). A stamp attached to a piston exerted a periodic step-like transversal force $F$ onto the UFPTs in three different setups, namely, for the rigid flat glass carrier and for the elastic silicone rubber carrier in flat or in pre-bent curved shape, which resulted in three different excitation schemes. **b** The charge response of a single transducer layer attached to the flat glass carrier for transversal peak loads ranging between 0.25 N and 10.25 N (left) and to the flat silicone rubber carrier for peak transversal loads ranging between 0.25 N and 1.25 N (right). The small baseline fluctuation is stemming from charges generated by thermal fluctuations. **c** The charge response of the transducer for glass and silicone rubber carriers from (**b**), plotted as a function of the applied force and pressure differences $\Delta F$ and $\Delta p$, respectively. From the strictly linear relation $\Delta Q$ ($\Delta F$) (and also $\Delta Q(\Delta p)$), a sensitivity value $S_F$ can extract as the slope of the regression line (generated by linear least square fits). The results of the FEM simulations of the charge response to transversal loading for UFPTs with three different parylene substrate thicknesses $D_{sub}$ on rubber carriers are shown on the right. **d** Charge response of one, two and three piezoelectric transducer layers attached to a pre-bent (curved) rubber carrier under repetitive transversal loading of $\Delta F = 2.5$ N.

($\varepsilon_{r, \text{ after poling}} < \varepsilon_{r, \text{ before poling}}$) is a clear indication of a high degree of crystallinity. Accordingly, we expect to see a close correlation between the change in relative permittivity $\Delta\varepsilon_r$, defined as $\Delta\varepsilon_r = \varepsilon_{r, \text{ after poling}} - \varepsilon_{r, \text{ before poling}}$, and $P_r$, which we indeed observed (Supplementary Fig. 4d).

As shown schematically in Fig. 3a, we can easily peel off the 1-µm thin UFPT films from the supporting glass carrier used upon fabrication and apply them to various other carriers with different shapes or material properties. To investigate the piezoelectric sensing and energy-generating properties of the UFPTs under the conditions of external mechanical stress, we performed transversal loading and bending tests, whereby the UFPTs were

transferred to elastic and rigid carriers (Fig. 3a). The rigid carrier was a 1-mm-thick glass sheet (with an elastic modulus $Y$ ~70 GPa) and the elastic carrier was either a 2-mm or a 6-mm-thick silicone rubber sheet ($Y$ ~1.45 MPa).

The ability of piezoelectric materials to generate sensor response or electrical power from mechanical deformations originates from the direct piezoelectric effect. External mechanical stress variations $d\sigma_{33}$ (here, tensile stress transversal to the film plane) and $d\sigma_{11}$ or $d\sigma_{22}$ (here, tensile stress longitudinal to the film plane), respectively, induce changes in the dipole density across the sample volume, and thus, they elicit a change in polarisation $P_3$. $P_3$ is the macroscopic polarisation that was

inscribed in the material during poling; it is oriented vertical to the electrodes. The deformation-induced polarisation change $dP_3$ results in fluctuations in the charge density at the electrodes, which can be measured as a piezoelectric current $I$ under short-circuit conditions.

Transversal-load tests were performed with a Shimadzu EZ-SX (1–500 N) with customised equipment as shown in Supplementary Fig. 6a (more detailed information is provided in "Methods"). The samples on glass and silicone rubber carriers were loaded with trapezoid-shaped step forces $F(t)$ with maximum levels that ranged from 0.25 to 10.25 N, as shown in Fig. 3b and Supplementary Fig. 6b, c. The charge response $Q(t)$ was obtained by numerically integrating the directly measured short-circuit current response $I(t)$. This response closely mimicked the shape of the applied force profile and allowed us to perform static force measurements. By examining the linear dependence of charge $Q$ on the force changes $\Delta F$ (Fig. 3c), we could determine a force sensitivity $S_F$ as

$$S_F = \Delta Q / \Delta F \qquad (1)$$

In the context of energy harvesting, $S_F$ corresponds to the charge that can be generated from a dynamically applied force. As shown in Fig. 3c and summarised in Supplementary Table 1, the force sensitivity for ultrathin transducers on glass is $S_F = 54$ ($\pm 8$) $pC\,N^{-1}$ within a force range of $0.1 < \Delta F < 10\,N$ (equivalent to a pressure sensitivity of $S_p = 11\,pC\,kPa^{-1}$ within a pressure range of $1 < \Delta p < 50\,kPa$). On silicone rubber carriers, we obtain $S_F \sim 1300\,pC\,N^{-1}$ for $0.1 < \Delta F < 1\,N$ (equivalent to $S_p = 260\,pC\,kPa^{-1}$ for $0.5 < \Delta p < 5\,kPa$). The sensitivity of samples attached to the more elastic silicone rubber carrier was more than 24 times higher than the sensitivity of samples on the rigid glass carrier, which indicates that elastic carriers strongly enhance the response signals for a given transversal load.

We could trace the origin of the different sensitivities on rigid and elastic carriers back to the different contributions of the stress/strain components. For the ferroelectric co-polymer, the piezoelectric change in the polarisation is directly related to the strain in the poling direction, which, for the UFPT, is the transversal direction[54]. On a rigid carrier-like glass, only the transversal stress (stress in the thickness direction) causes a transversal strain and contributes to the transducer response, as almost no longitudinal strain appears because of clamping by the rigid carrier. Elastic carriers will yield to the applied stress, which causes a strong deformation, especially close to the edge of the stamp. In this region, the deformation of the carrier induces a large longitudinal strain in the ferroelectric layer, which in turn transforms into a transversal strain by transverse contraction, as shown in FEM simulations (Supplementary Fig. 7). This indirectly caused transversal contraction in the vicinity of the stamp edge surpasses by far the transversal strain values in the ferroelectric layer below the flat contact region with the stamp (Supplementary Fig. 7b). The FEM simulation predicts that the charge response on the same elastic carrier increases when the thickness of the substrate UFPT (parylene), $D_{sub}$, decreases, as can be seen in the right plot in Fig. 3c. Note that the quantitative agreement between the measured and simulated charge response for the 1-µm thin parylene substrate is remarkable (red and blue dots in the left plot of Fig. 3c).

The ultraflexibility of our UFPTs makes it easy to stack or roll transducers. This offers unique advantages, thereby enabling us to optimise the total power output and adjust the overall impedance. As shown in the response curves in Supplementary Fig. 8, stacking two piezoelectric transducers on a 6-mm-thick silicone rubber carrier can double the charge response under a transversal pressure load, thereby yielding a pressure sensitivity of $S_p = 520\,pC\,kPa^{-1}$ in the range of $0.5 < \Delta p < 5\,kPa$ (compare with Supplementary Table 1). Further, we vertically attached three layers of ultraflexible transducers to each other and mounted them on a curved/bent soft carrier (2-mm-thick silicone rubber) as shown in Supplementary Fig. 9, allowing us to integrate the components in a perfectly conformable way. At a transversal load of 2.5 N, the generated charge level was either doubled for the two-layer stack or even tripled for the three-layer stack as compared to the maximum charge level measured for a single-layer UFPT under the same excitation conditions (Fig. 3d). These values are presented in Supplementary Table 1, thereby providing an overview of the most important performance parameters for our single- and multilayer UFPTs.

The UFPTs exhibit excellent mechanical stability. Bending tests were performed by conformably attaching the transducer to a thin gold wire with a diameter of 80 µm, as displayed in the inset of Fig. 4a. The perfect overlay of the hysteresis curves $D(E)$ recorded before bending, during bending and after bending clearly shows that the ferroelectric properties of the UFPTs were not influenced at all by bending along such a small radius of 40 µm (Fig. 4a).

Moreover, mechanical fatigue tests (Fig. 4b, c) did not reveal any systematic change in the current response of the transducer (no data distortion, no fluctuation) during repetitive longitudinal (strain/release) and transversal (push/release) loadings. When the transducers were actuated over more than 1000 (longitudinal)/ 6000 (transversal) cycles, the generated current response remained perfectly reliable and stable. The measurement setup, as well as the current and charge response curves upon tensile strain tests, is displayed in Supplementary Fig. 10. Another big advantage of piezoelectric transducers (and thus PENGs) is their rapid response. As demonstrated in Fig. 4d, the charge response of the transducer nearly instantaneously followed the applied force profile revealing a rise time well below $20\,ms\,N^{-1}$.

**Wireless healthcare-monitoring device (E-health patch).** In order to create a real-time healthcare-monitoring device that is conformable enough to the skin surface to be attached directly and without adhesive, the UFPTs were combined with a small compact wireless module. This e-health patch device can monitor vital signs such as pulse rate, human pulse wave, and respiratory rate. A photograph of the wireless patch in operation is displayed in Fig. 5a, which shows a virtually imperceptible neck-mounted sensor (~2 mg without wiring) connected to a very compact and lightweight (~5.6 g) wireless module. This in turn was attached to the skin beneath the collarbone. Since the transducer is ultra-flexible and the electronic module is lightweight, our e-health patch is comfortable to wear.

This setup was used to monitor the human pulse wave in real time and determine the rate of the pulse wave in a 32-year-old woman at rest (Fig. 5a and Supplementary Movie). We could extract a pulse rate of 54 $min^{-1}$ and an artery augmentation index (AI) of 56% from the recorded and wirelessly transmitted data. The AI value $(AI\,(\%) = P_2/P_1 \cdot 100)$ was determined from the shape of the human pulse wave, which is shown in the enlargement of the recorded signal in Fig. 5a. The AI correlates with the elasticity of the human blood vessel; a calculated value of AI = 56% indicates a healthy 32-year-old women[55]. In addition, we used this setup to monitor the pulse wave on the wrist, whereby a pulse rate of 60 $min^{-1}$ was extracted (see Supplementary Fig. 11).

By measuring the pulse wave with two or more sensors placed a certain distance from one another, the pulse wave velocity (PWV) could be determined. The PWV allowed us to estimate the human blood pressure. Figure 5b shows pulse wave measurements from two ultraflexible ferroelectric sensors attached to the neck of a 34-year-old man. By examining the propagation delay $\Delta t$ of the

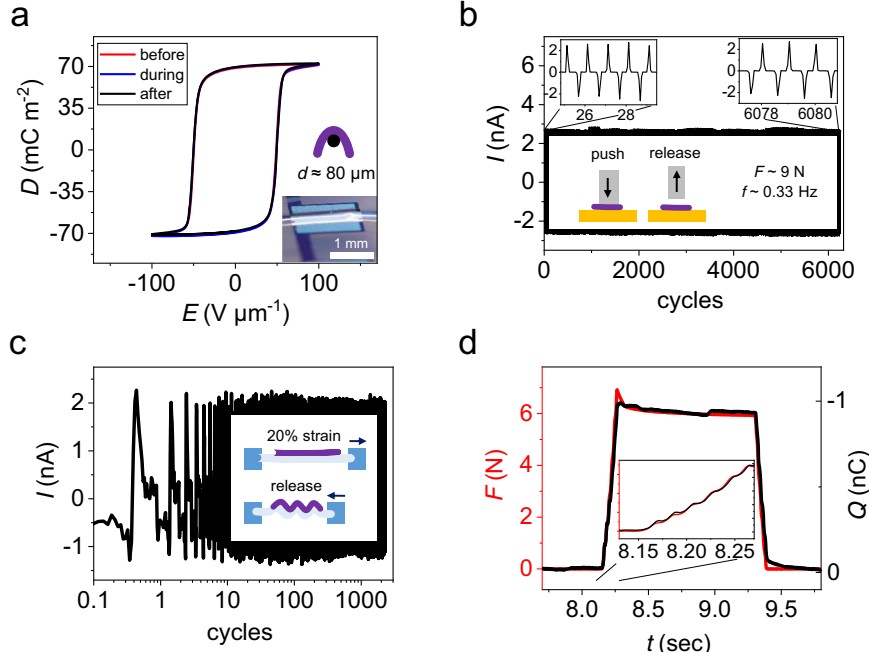

**Fig. 4 Stability test and response time measurements of UFPTs. a** $D(E)$-hysteresis curve of an UFPT before, during and after bending over an 80-µm-thick gold wire. Durability testing under (**b**) transversal-load and (**c**) longitudinal-strain conditions. **b** Current response when a transducer mounted on a glass carrier is subjected to repeated transversal compression and release via a stamp over a period of more than 5 h (>6000 cycles). **c** Current response upon longitudinal strain cycling over more than 1000 periods. The inset schematically illustrates the longitudinal tensile test procedure. First, the UFPT was mounted on a 20% pre-strained rubber carrier, which was clamped at both ends. Then, by periodically relaxing and stretching the carrier (light blue) over an interval of 0–20% strain, the transducer (purple) was contracted or retracted, respectively. **d** Time dependence of the charge response of the UFPT for a trapezoidal transversal load with a top force level of 6 N and a rise time of « 20 ms N$^{-1}$. The charge response signal (black) precisely follows the force profile (red).

signal and the distance between the sensors $\Delta x$, a PWV of 9 m s$^{-1}$ was calculated. As suggested by Ma et al.[56], the human blood pressure $P_B$ (in the range of 5–20 kPa) can be estimated from the PWV as

$$P_B = \alpha \cdot \mathrm{PWV}^2 + \beta; \quad \alpha = 0.18\,\mathrm{kPa \cdot s^2 \cdot m^{-2}}; \quad \beta = 2.7\,\mathrm{kPa}, \quad (2)$$

yielding a blood pressure $P_B$ of 130 mm Hg for the 34-year old male. In Eq. (2), the human arteries are characterised using the Fung hyperelastic mode; model (2) is validated by data from the literature and from experiments conducted on human subjects[56].

The authors are aware that there exist many different sensor technologies to measure the human pulse wave, ranging from optic[57,58] over ultrasonic[59] to force[20,54,55,60–65] sensing approaches. The force-sensing approaches make use of piezoresistive[60,61], piezoelectric[54,55] or triboelectric[62] effects, or of capacitive changes[20,63–65]. Many of those devices are impressive with respect to their high sensitivity and ultrafast response time; for instance, Yao et al. recently reported a piezoresistive sensor with an impressive sensitivity >10$^7$ Ω kPa$^{-1}$ and a fast response time of 1.6 ms[60]. However, only a few can combine high sensitivity and fast response time with low power consumption, flexibility/conformability and biocompatibility.

The UFPT sensor technology excels for pulse wave monitoring in that it combines many aspects: it is self-powered (charge generation, not consumption), shows excellent mechanical stability (>1000 loading cycles), and has a high sensitivity (>10$^3$ pC N$^{-1}$) while offering ultrafast response (« 20 ms N$^{-1}$). Furthermore, its ultraflexibility enables conformal attachment to various materials and surfaces as well as multilayer stacking even on 3D-shaped carriers for further improvement in sensitivity (>10$^4$ pC N$^{-1}$).

**Comprehensive ultraflexible energy-harvesting device.** In the ultraflexible energy harvesting devices (UEHDs), useable to power the electronic module of the e-health patch, the ultraflexible transducers were utilised as piezoelectric nanogenerators. Since for most mechanical stimuli (e.g., vibrations or arm/knee bending) the piezoelectric transducers generate an alternating current (AC) signal, a rectifier circuit is needed to convert the AC signal into a direct current (DC) signal, which can then be used to charge energy-storage devices such as capacitors or rechargeable batteries. We constructed the rectifier from organic diodes, whereby two different diode architectures were tested: (i) a vertical Schottky diode architecture, as shown in Supplementary Fig. 12 and (ii) an organic thin-film transistor (OTFT)-based architecture with shorted gate-drain electrodes, as depicted in Fig. 6a. The most important diode performance parameters of each setup are listed in Supplementary Table 3.

Since the Schottky diode architecture had several disadvantages, such as a substantial built-in voltage drop, a worse device-to-device reproducibility and limited electrical stability (for more information please refer to Supplementary Note 2), we focused on the OTFT-based architecture. Diodes utilising the organic semiconductor DNTT were fabricated in a bottom-gate top-contact OTFT layer setup where the gate and drain contacts were connected and thus short-circuited through vias (Fig. 6a). As a gate dielectric 10-nm-thick anodised alumina with a SAM (12-dodecylphosphonic acid, C$_{12}$-PA) modification layer was used, which had already demonstrated excellent dielectric and interfacial properties in high-performance DNTT-OTFTs[66]. We fabricated these shorted OTFT diodes on parylene substrate with only 1-µm thickness thus forming ultraflexible and lightweight organic diodes never reported before.

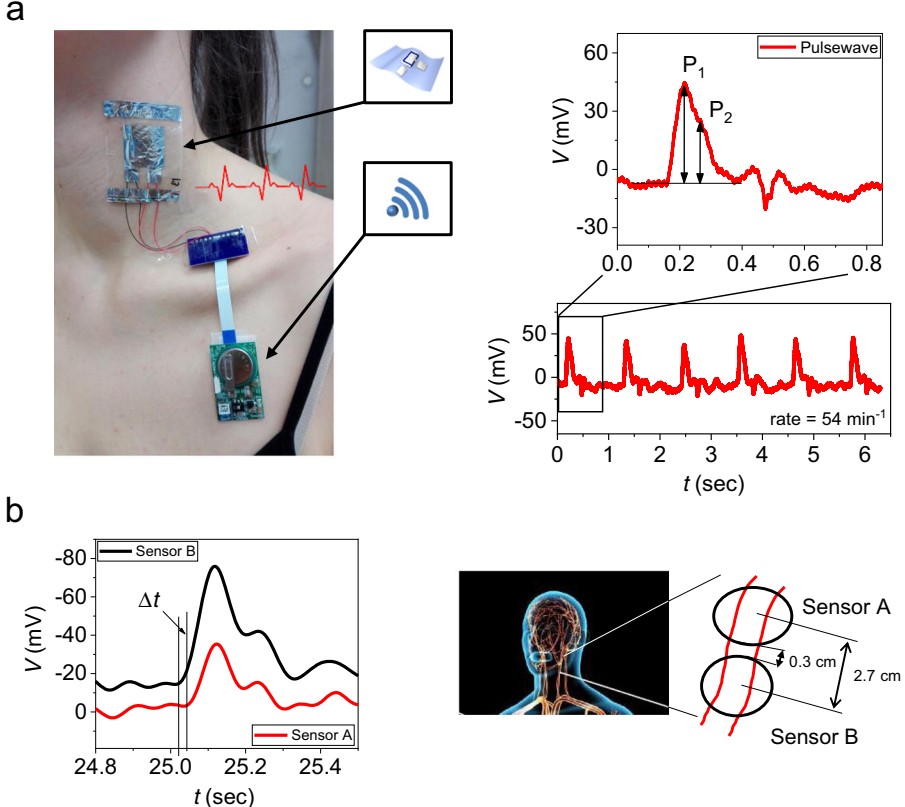

**Fig. 5 Wireless e-health patch.** Attachment points of the wireless e-health patch, whereby the ultraflexible transducer serves as an imperceptible sensor that adheres to the skin without adhesive. The patch can monitor (**a**) the human pulse wave (with $P_1$ and $P_2$ peaks) from which the rate (here: 54 min$^{-1}$ for a 32-year-old woman) and the artery augmentation index AI (here: AI ~56%) can be measured as well as (**b**) the blood pressure of the human arteria in the neck via the pulse wave velocity PWV. PWV can be determined by measuring the signal delay $\Delta t$ for a given sensor distance $\Delta x$. In this example, PWV was ~9 m s$^{-1}$ for a 34-year-old man.

Typical current/voltage characteristics of the OTFT-based diodes for three different $W/L$ ratios (500/12, 7000/12, and 27,000/12, all in μm) are plotted in Fig. 6b. We observed excellent rectification ratios of up to $10^7$, transition voltages $V_T < 0.1$ V, reverse breakdown voltages $V_{break} < -5$ V, and current densities $J$ ~65–105 mA cm$^{-2}$ at a forward voltage of only $V = 2$ V. The corresponding transistor transfer characteristics (without gate/drain shortage) for the highest $W/L$ ratio (27 mm/12 μm) are plotted in Fig. 6c (left, black curve). We extracted a threshold voltage of $V_T = -0.5$ V, an onset voltage close to $V_{on} = 0$ V, a charge carrier mobility of $\mu = 0.6$ cm$^2$ V$^{-1}$ s$^{-1}$, a subthreshold swing $S < 100$ mV and an ON/OFF ratio above $10^7$. From the transistor transfer curve, we could very well estimate the performance of the organic diode with shorted gate-drain electrodes, as displayed in Fig. 6c. The black curve of the diode $I(V)$ characteristics (Fig. 6c, right plot) corresponds to the black curve of the OTFT transfer characteristics (Fig. 6c, left plot). The measured onset voltage of the transistor transfer characteristics has a strong relation to the diode performance as illustrated by two other diodes $I(V)$ curves plotted in red and blue. A highly negative transistor onset voltage would yield a positive transition voltage in the diode (blue curve in Fig. 6c, right) resulting in a parasitic voltage drop across the diode. That would decrease its rectifying performance. In contrast, a transistor onset that is overly positive (transistor transfer curve in red in Fig. 6c, left) increased the diode's off-current (red curve in Fig. 6c, right), which again strongly reduced its rectification ratio. To ensure optimum diode performance regarding rectification, the transistor should exhibit an onset voltage that is close to zero or slightly

negative and should have a low subthreshold swing, low gate leakage current, and high charge carrier mobility.

The OTFT-based diode performance can be improved by increasing the $W/L$ ratio; namely, the rectifying ratio was increased from $10^5$ ($W/L$: 0.5 mm/12 μm) to over $10^7$ ($W/L$: 27 mm/12 μm) accompanied by a decrease of the output impedance (see also Supplementary Table 3).

This is illustrated in Supplementary Fig. 13c, where the rectifying ratios of 31 OTFT-based diodes for three different $W/L$ ratios (27 mm/12 μm, 7 mm/12 μm, 0.5 mm/12 μm) are plotted as a histogram. One can clearly see that the average rectifying ratio increases with increasing $W/L$.

A photograph and equivalent circuit of an organic full-wave rectifier (OFWR) made of four OTFT-based diodes with $W = 7$ mm are displayed in Fig. 6d. The OFWR circuit could rectify both half cycles of a sine wave AC input signal ($V_{in}$) if it was connected to a 1-MΩ resistor, as demonstrated in Supplementary Fig. 13a, whereby the maximum amplitude of the input signal of 2 V was slightly reduced to 1.8 V. Coupling the as-rectified signal to a capacitor of $C = 10$ μF resulted in a smoothened DC output voltage $V_{dc}$ of about 1.6 V (Fig. 6e). The OFWR circuits exhibited excellent electrical stability. We tested that 2.5 h of continuous operation reduced its DC output by only 5% (Supplementary Fig. 13b). This small reduction occurred because of a 'reversible' bias stress effect that caused a negative shift in the transition voltage of the organic diode (induced by charge trapping at the dielectric/semiconductor interface). This, in turn, resulted in a small decrease in the rectifying performance. After the measurement, the negative shift in the transition voltage stabilised within

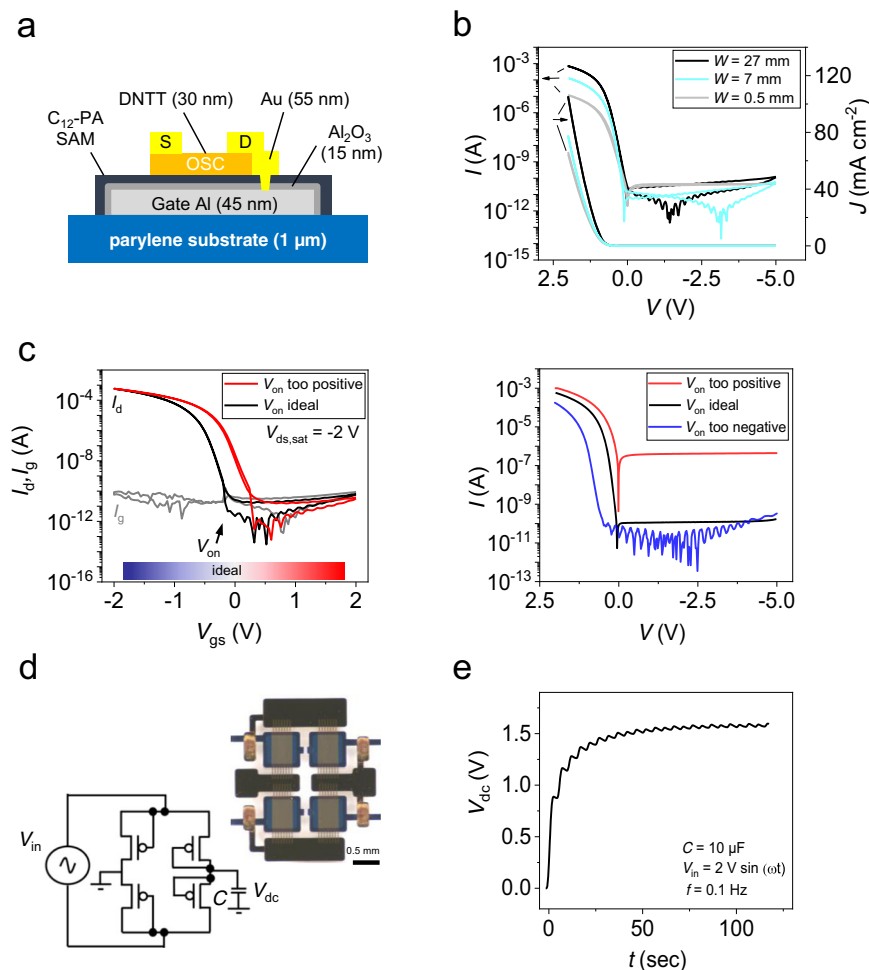

**Fig. 6 Ultraflexible organic rectifier circuit. a** Scheme of the organic rectifier diode created with an OTFT by shortening the drain and gate contacts ($C_{12}$-PA = 12-dodecylphosphonic acid, OSC = organic semiconductor, S = source and D = drain). **b** Representative $I/V$ ($J/V$) curves of the organic diode for different channel widths $W$ (channel length $L$ fixed to 12 μm) fabricated on 1-μm thin parylene. $J$ is the current density. **c** The electrical transfer characteristics of the OTFT (left plot) are related to its performance when used as a diode (right plot) by shortening the drain-gate contacts. The characteristics in the left plot correspond to the black and red graph ('$V_{on}$ ideal' and '$V_{on}$ too positive') in the right plot (see main text). **d** Photograph and equivalent circuit of an OTFT-based full-wave rectifier circuit (OFWR) with $W/L = 7000$ μm/12 μm fabricated on 1-μm thin parylene. **e** Rectified output signal from an OFWR fed by an AC input signal $V_{in}$ (2 V sin ($2\pi f \cdot t$), $f = 0.1$ Hz) and connected to a capacitor of $C = 10$ μF.

a few minutes, and the DC output level ($V_{dc}$) recovered to its initial value.

Supplementary Fig. 14 shows the comprehensive equivalent circuit for the harvesting of biomechanical energy by means of the ultraflexible piezoelectric energy-harvesting device (UEHD). As already described, the UEHD comprises a piezoelectric nanogenerator (represented here by a single or stacked ultraflexible ferroelectric polymer transducer), a rectifier circuit (represented here by an organic full-wave rectifier), an energy-storage device (represented here by a thin-film capacitor) and a load (represented here by an LED).

One of the most efficient ways to induce mechanical excitations is by bending the piezoelectric material. Since we can achieve especially small bend radii in UFPTs (see Fig. 4a), we expect an excellent energy-harvesting performance. We tested three different bending modes: (A) continuous bending by hand with results shown in Fig. 7b and Supplementary Fig. 15; (B) controlled continuous bending over a rail, see Fig. 7 and Supplementary Fig. 16 and (C) continuous pressing with the fingertip on the bent transducer, see Fig. 7b, d and Supplementary Fig. 17. Please refer to the "Methods" for more detailed information about bending tests.

The power output parameters for the three different harvesting modes are listed in Table 1. For the manual bending (mode A) performed at ~2 Hz, an open-circuit voltage with a peak value of about $V_{oc} = 5.5$ V and a current peak under short-circuit conditions of $I_{sc}$ ~0.9 μA were measured (see Supplementary Fig. 15a for the time plot). By varying the external resistance, the maximum volumetric output power density reached $P_{out,max}$ ~3.2 mW cm$^{-3}$ (corresponding to an areal power density of $P_{out,max}$ ~0.8 μW cm$^{-2}$) for an optimum load resistance $R_L = 5$ MΩ (Fig. 7b), which is among the highest values ever reported for PENGs[1,4].

By forward and backward movement over a rail at 2 Hz (mode B), the PENG is subdued to continuous bending and releasing and thereby we could harvest the energy in a much better controlled manner. Since the current responses of a PENG to bend and release events have opposite signs, they need to be rectified—otherwise, the signals would cancel each other out. With the OFWR we could efficiently convert the generated AC to DC signals and finally store them in a capacitor. The corresponding charging curve of the capacitor is shown in Fig. 7c. The enlargement demonstrates that both the energy generated during the bending motion (highlighted blue) and that generated

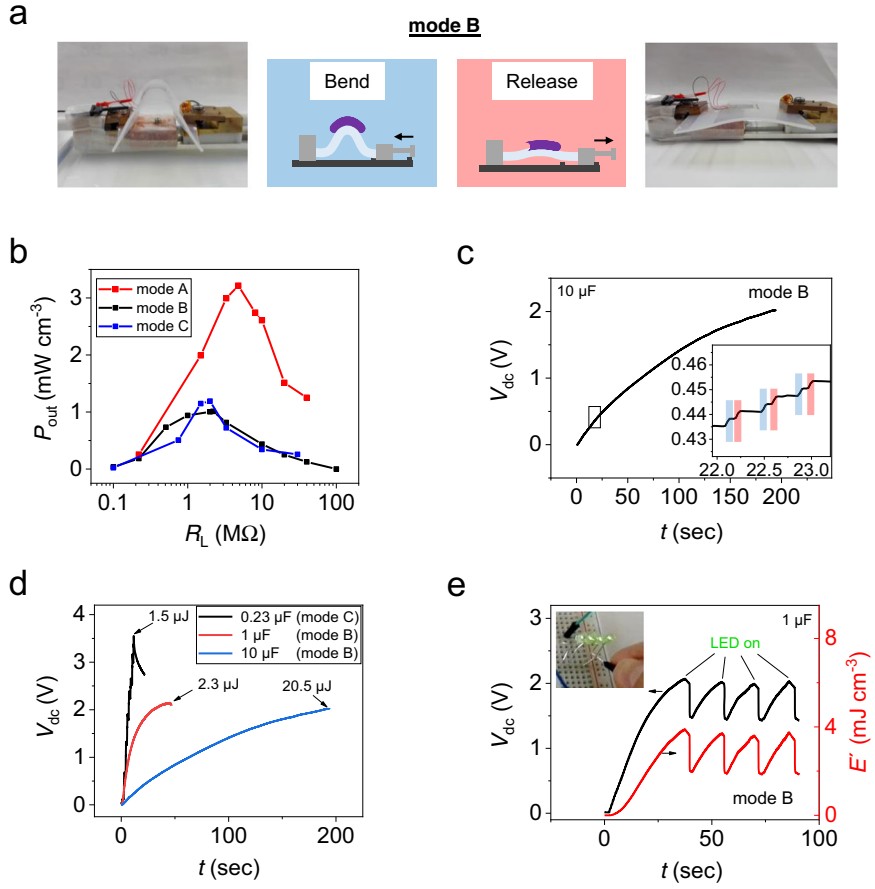

**Fig. 7 Energy harvesting.** Harvesting of actuation energy generated by periodic (2 Hz) bending and releasing of an UFPT placed on top of a 2-mm-thick rubber layer via sliding on a rail (mode B). **a** Photographs and schematics of the bending procedure. **b** The output power density $P_{out}$ of mode B is plotted as a function of load resistance $R_L$ (black curve) and compared to those from mode A (red curve) and mode C (blue curve). **c** Charging curve of a capacitor for an UFPT excited in mode B. The UFPT is connected to the ultrathin OFWR, which then charges the capacitor ($C = 10 \,\mu F$). The enlargement shows the generated energy steps related to bending (blue) and releasing (red) motions. **d** The capacitor voltages $V_{dc}$ plotted over time and the maximum stored energy levels ($E = \frac{1}{2}C \cdot V_{dc}^2$) are shown for three different capacitor values, whereby the smallest capacitor with 0.23 μF is an ultraflexible thin-film capacitor fabricated on the 1-μm thin parylene substrate. For charging the thin-film capacitor, the UFPT was excited by bending mode C. Capacitors were charged almost to saturation. The maximum charging voltage depends on the generated energy of the PENGs (voltage levels), discharging effects and parasitic voltage drops along the energy harvesting circuits. **e** The time dependence of capacitor voltage $V_{dc}$ and of the energy density $E'$ generated by periodically bending an UFPT and dissipating the energy by powering four LEDs that are connected in parallel.

### Table 1 Power output of UEHD for different energy-harvesting modes.

| Mode[a] | Setup[b] | $I_{sc}$ (μA)[c] | $V_{oc}$ (V)[d] | $P_{out,max}$ (mW cm$^{-3}$)[e] | $<P_{out}>$ (mW cm$^{-3}$)[f] |
|---|---|---|---|---|---|
| (A) Manual bending | UFPT on silicone rubber | ~0.9 | ~5.5 | ~3.2 (5 MΩ) | n/a |
| (B) Controlled bending | UFPT on bent silicone rubber (fixed on rail) | ~3.0 | ~2.0 | ~1.0 (2.5 MΩ) | ~0.1–0.2 |
| (C) Pressing with fingertip | UFPT on bent silicone rubber (fixed on a rail) | ~2.0 | ~3.5 | ~1.2 (2 MΩ) | ~0.25 |

[a]Operation mode; [b]layer setup used; [c]$I_{sc}$: short-circuit current; [d]$V_{oc}$: open-circuit voltage; [e]measured output power density $P_{out}$ for optimum load resistance (denoted in bracket) at ~2 Hz excitation frequency; the active UFPT volume is $A \cdot D_{trans} = 5.63 \times 10^{-4}$ cm$^3$, whereby $A$ is the active transducer area of 2.25 cm$^2$ and $D_{trans}$ is the total transducer thickness of 2.5 μm; [f]the mean generated power density $<P_{out}>$ is calculated by $P = E \, \Delta t^{-1}$, whereby $E$ is the harvested energy stored in the capacitor $C$ and is defined by $E = \frac{1}{2} C V_{dc}^2$.

during the releasing motion (highlighted red) were harvested. This is proof that the OFWR operated well. The dependence of the output power $P_{out}$ on the load resistance $R_L$ is plotted in Fig. 7b, with $P_{out,max}$ ~1.0 mW cm$^{-3}$ (0.25 μW cm$^{-2}$) for a load resistance of $R_L = 2.5$ MΩ. In Fig. 7d, the charging curves are displayed as a function of harvesting time (bending events) for three different capacitor values. After a single PENG was bent 20, 100 and 450 times, energies equivalent to $E = 1.5$, 2.3 and 20.5 μJ could be stored in capacitors with $C = 0.23$, 1 and 10 μF, which correspond to charging voltages of $V_{dc,max} = 3.5$, 2.15 and 2 V,

respectively. We must emphasise that the small capacitor ($C = 0.23$ μF) is based on an ultraflexible thin-film system made of an thin $Al_2O_3$ layer (10 nm) sandwiched between Al electrodes fabricated on the 1-μm thin parylene substrate and for charging the thin-film capacitor, the UFPT was excited by bending mode C. The layer setup and the $C(f)$-characteristics of this ultraflexible capacitor are displayed in Supplementary Fig. 18. The mean generated power $<P_{out}>$ ranged between 0.1 and 0.25 mW cm$^{-3}$, which was sufficient to charge a 1 μF capacitor within 35 s (70 bending steps) to 2 V and power four LED lamps connected

in parallel (RoHs: $V_f = 1.9$–$2.4\,V$, $\lambda = 570\,nm$), as demonstrated in Fig. 7e.

From the energy levels generated during bending motions (mode B), we roughly estimated that more than 200 mJ per day can be gained from biomechanical motions if multilayer UFPTs are placed on joints like knees or elbows. For a detailed description, please be referred to Supplementary Note 3.

By continuous pressing with the fingertip on the bent transducer (mode C) a maximum power $P_{out,max}$ ~1.2 mW cm$^{-3}$ (0.3 µW cm$^{-2}$) for a load resistance of $R_L = 2\,M\Omega$ (Fig. 7b and Supplementary Fig. 17b) was observed which is lower in comparison to bending mode A (manual bending: $P_{out,max}$ ~3.2 mW cm$^{-3}$), but interestingly quite similar to bending mode B (bending on rail: $P_{out,max}$ ~1.0 mW cm$^{-3}$).

We further demonstrated that the energy output can be increased proportionally by stacking multiple PENGs onto a pre-bent rubber carrier and periodically touching them with a finger (mode C, Supplementary Fig. 17a). For a triple-stack PENG, it was possible to generate more than three times as much energy than could be produced with a single-layer PENG (Supplementary Fig. 17c).

A vital parameter in terms of energy harvesting is the efficiency of conversion from mechanical to electrical energy[23]. The total mechanical input energy needed to cause actuation of the harvester can hardly be determined as it strongly depends on the body the harvester is attached to in terms of its mechanical properties, dimension, deformation shape and other quantities. However, one can define the energy-conversion efficiency as the ratio of harvested electrical energy to the stored strain energy upon deformation of the whole harvester system, including the passive substrate, i.e.,

$$\eta = \frac{W_t^{el}}{W_t^m} \cdot \frac{W_t^m}{W_{sub}^m + W_t^m} \qquad (3)$$

where $W_t^{el}$ / $W_t^m$ is the conversion ratio of mechanical strain energy ($W_t^m$) to electrical energy ($W_t^{el}$) given by the piezoelectric material, and $W_{sub}^m$ is the mechanical energy stored in the passive substrate. We performed 3D FEM simulations of the bending experiment (mode B) to estimate the energy levels for different substrate thicknesses in this actuation mode and compared the results with an analytical model (see Supplementary Note 4). Obviously, a large strain energy ratio $W_t^m$ / $W_{sub}^m$ is essential for a high conversion efficiency, which can be approximated by

$$\frac{W_t^m}{W_{sub}^m} \approx \frac{D_t}{D_{sub}} \cdot \frac{E_t}{E_{sub}} \cdot \frac{1 - \nu_{sub}^2}{1 - \nu_t^2} \qquad (4)$$

with $E_t$, $\nu_t$ and $E_{sub}$, $\nu_{sub}$ being Young's modulus and Poisson's ratio of the piezoelectric and the substrate layer, respectively, and $D_t$, $D_{sub}$ denoting the respective layer thicknesses (cf. Supplementary Fig. 20a). The energy ratio thus scales inverse with the product $D_{sub} \cdot E_{sub}$. For the presented UFTP in bending actuation, the model given by Eqs. (3) and (4) predicts a conversion efficiency of $\eta = 0.185\%$, whereas the more accurate 3D simulation gives $\eta = 0.139\%$. From Supplementary Fig. 20b, it is clear that a very small substrate layer thickness is significant to obtain a high-energy-conversion efficiency. When using a ten times thicker substrate, i.e., 10 µm, the simulated efficiency significantly drops to only 0.018%, which is more than seven times lower, and for a 100-µm-thick substrate, it is even 25 times lower. This highlights the major improvement in energy conversion of P(VDF-TrFE)-based transducers by drastically decreasing their substrate thickness and thus strongly supports the concept of ultraflexibility.

## Discussion

In this study, we demonstrated energy harvesting and sensor devices with ultraflexibility like an ultraflexible ferroelectric polymer transducer, an ultraflexible organic diode and ultra-flexible rectifier circuits. All devices were fabricated on only 1-µm thin parylene substrate and have an overall thickness <2.5 µm.

For the ultraflexible transducers, we investigated the dielectric, ferroelectric and crystallographic properties of the P(VDF: TrFE)$_{70:30}$ co-polymer layer as a function of the annealing temperature. XRD measurements showed that annealing at 130 °C induced a highly crystalline (84%) β-phase with a large remnant polarisation of $P_r = 67\,mC\,m^{-2}$ resulting in a high transversal force sensitivity up to 1.3 nC N$^{-1}$ for a single-layer UFPT attached to a silicone rubber carrier. We noted that the sensitivity strongly depends on the deformability of the underlying carrier. The ultrathin UFPT imitates the complex three-dimensional deformations of the underlying material perfectly, which can positively add to the overall piezoelectric response and thus lead to increased sensitivity. Furthermore, the FEM simulation results reveal that the ultrathin parylene substrate is responsible for the high degree of force and pressure sensitivity and boosts the energy-conversion efficiency. We can attach the UFPTs to different materials and surfaces of almost any 3D shape in a straightforward way. Owing to their ultraflexible and good adhesion properties, they can be easily stacked, thereby multiplying the respective sensing/harvesting performance proportionally. We achieved transversal force sensitivities up 15 nC N$^{-1}$ for a triple UFPT stack on a 3D-bent elastic rubber surface. The UFPTs showed excellent mechanical stability with a bending radius down to 40 µm and an ultrashort response time «20 ms N$^{-1}$. For biomedical signal recording they were integrated as imperceptible sensors in a wireless healthcare patch to allow real-time precise monitoring of heart rate, AI-index, and even blood pressure in a cuff-free, non-invasive way. Our imperceptible e-health patch is thus capable of detecting lifestyle-related diseases such as heart disorders, signs of stress and sleep apnoea, early on.

In order to allow for piezoelectric energy harvesting from periodic or repetitive mechanic excitation schemes like vibrations, joint movements or breathing, the periodic response signals have to be rectified to avoid cancellation. We developed ultraflexible organic full-wave rectifier circuits (OFWR) constructed of four organic diodes in a transistor architecture with shorted gate-drain contacts. The OTFT-based diodes exhibited an excellent rectifying ratio of up to 10$^7$, very-low-transition voltages <0.1 V and high bias stress stability. These organic diodes also excel in terms of flexibility.

Deployed as piezoelectric nanogenerators the UFPTs could deliver peak power densities of over 3 mW cm$^{-3}$ (0.75 µW cm$^{-2}$) upon bending. By combining the PENGs with the OFWR circuit and ultraflexible thin-film capacitors, we formed an ultraflexible and thus imperceptible energy-harvesting device with an average continuous output power density up to 0.25 mW cm$^{-3}$, even at very-low excitation frequencies (~2 Hz).

With regard to power supply for e-health patches, we estimate that energy levels of more than 200 mJ per day can be reached by energy harnessing from biomechanical motions if multilayer UFPTs are placed on joints, like knees or elbows. This is sufficient to power a wireless electronic system operating in an ultra-low power consuming duty-cycled fashion[67] and should allow to transmit the measured pulse wave data several times a day (e.g., once or twice an hour). Yet, these values are based on several assumptions, thus further research and extended field tests are necessary to test the long-term energy harvesting potential of our technology on different parts of the body.

## Methods

**Ultraflexible substrates**. As substrate, a 1-μm-thick parylene diX-SR (from Daisan Kasei Co., Ltd.) was formed by chemical vapour deposition on a fluoropolymer-coated glass supporting glass carrier. This setup was used to fabricate ferroelectric transducers, organic rectifying circuits, and thin-film capacitors. The fluoropolymer CYTOP™ (CTL809M mixed with CT-Solv.180 in a ratio 1:6) from AGC coated on a supporting glass carrier allowed the devices to be peeled off surfaces after all fabrication steps.

**P(VDF:TrFE) transducer fabrication**. The ferroelectric transducers were fabricated in a capacitor-like structure with a spin-coated ferroelectric layer (10 wt% P (VDF:TrFE)$_{70:30}$ in γ-butyrolactone, 2000–3000 rpm; solutions for spin coating were prepared from P(VDF:TrFE) powders as purchased from Piezotech, Arkema) sandwiched between two metal layers (Fig. 2a). Bottom and top electrodes were formed by the thermal evaporation of a 90–110-nm-thick aluminium layer through a shadow mask at a rate of 1–2 nm s$^{-1}$ in a high vacuum system at a pressure of 10$^{-4}$ Pa. After the ferroelectric layer was spin-coated (MS-A1500 Opticool by Mikasa), samples were annealed for 5 min on a hot plate followed by 1 h of annealing in a vacuum oven at the same temperature as used on the hot plate. Temperatures ranged from RT to 170 °C (RT, 50 °C–170 °C in 20 °C steps). Directly after the annealing step, the samples were taken out of the vacuum oven and cooled down to RT.

**Electrical poling and hysteresis measurements of the ferroelectric P(VDF: TrFE) layers**. To achieve poling, a programmable signal generator (B2912A Precision Source/Measure Unit, 2 ch, 10 fA system) is used. Sinusoidal voltage waveforms with amplitudes up to 200 V and a frequency of 1 Hz are applied over several periods to the samples, while simultaneously recording the current flow, which allows constructing D–V hysteresis loops (Supplementary Fig. 1). To do so, the current signals were numerically integrated to obtain the charges at the electrodes of the same polarity $Q$, which can be interpreted as a compensation charge for a polarisation-induced surface charge forming at the interface between one electrode and the ferroelectric. The electric displacement $D$ thus gives rise to a surface charge density $\sigma_{el}$ at the electrode–ferroelectric interface, and it is directly related to the compensation charge (integrated measured poling current) in the electrode as

$$\sigma_{el} = Q_{el}/A_{el} \cong D, \qquad (5)$$

where $A_{el}$ is the contact area of the planar electrode. As the electrode geometry is known, one can therefore easily use the equation to plot displacement $D$ over the electric field $E$, with $E = V_{pol}\, d^{-1}$, to obtain a hysteresis plot. A maximum electric field about twice the coercive field strength ($\approx$100 V μm$^{-1}$) was applied to achieve sufficient poling of P(VDF:TrFE) films.

**Investigation of film thicknesses and morphology**. Film thicknesses and film morphologies were determined by profilometry (Dektak XT by Bruker) and atomic force microscopy (AFM5000 by Hitachi). The AFM images were analysed using the free WSxM software available from Nanotec Electronica[68]. The XRD measurements were performed to determine the crystallinity $X_c$ of P(VDF:TrFE) films using a Smartlab XPS system from Rigaku. Radiation with a wavelength of 1.4 Å and an incidence angle of 0.2° was selected for the primary beam (in-plane measurements). Data acquisition took place at a fixed sample position.

**Determination of crystallinity $X_c$**. The calculation of crystallinity by XRD is based on the presumption that the broad peak comes from the amorphous phase, whereas the sharp peak comes from the crystal phase. The diffraction curve around 20° is analogous to the β-phase crystalline plane of the P(VDF:TrFE)$_{70:30}$, whereas the shoulder is associated with the scattering from the non-crystalline regions (see Supplementary Fig. 5a). As a result, the diffraction curve observed can be resolved into two regions related to the crystalline and amorphous phases, respectively. By properly fitting the two peaks (superposing a Gaussian function to determine the integrated peak areas and refine the peak positions in the diffraction pattern (around 20°), the degree of crystallinity could be calculated as a ratio between crystal-related scattering intensity and the total scattering intensity (from crystalline + amorphous phases) according to

$$X_r = \frac{\sum A_{crystl.}}{\left(\sum A_{crystl.} + \sum A_{amorph}\right)} \qquad (6)$$

**Transversal load, tensile tests and bending test**. A Shimadzu EZ-SX (1–500 N) with customised equipment was used to perform transversal load and tensile tests. To enable connectivity of the transducer electrodes during these tests, thin isolated metal wires (PTFE AWG26, 0.31 mm, KYOWA) were connected with stretchable Ag-ink SX-ECA from CEMEDINE. The setup used to measure the transversal-load response and tensile strain response is depicted in Supplementary Figs. 6 and 10, respectively. To perform transversal pressure tests, exchangeable metal stamps were used to transfer the compression force over a contact area of 2 cm² to the transducer, while the force levels were monitored with a calibrated load cell. A cutout

PET film (Melinex ST506) on top of the transducer served as a pressure distribution layer and ensured that the contact area of the force was a bit smaller than the sensitive area. The force profile was trapezoidal and oscillated between 0.25 N and 10.25 N (Supplementary Fig. 6). As a bottom layer, glass or silicone rubber was used. Tensile tests were performed by first fixing the ultraflexible transducers with a Tegaderm™ film from 3 M on a pre-stretched silicone rubber carrier under 20% strain (photograph of the setup shown in Supplementary Fig. 10). Then, a tensile force oscillating between 1 N and 17 N was applied to the rubber layer, causing a strain variation from 0 to 20% in the rubber, respectively. The self-made setup for the bending test is displayed in Supplementary Fig. 16a. A 2-mm-thick silicone rubber film was fixed with two clamps at opposite ends, with the clamps being mounted on a rail. While the left-hand clamp was fixed, the right-hand one could be moved between two positions, thereby causing the rubber layer to bend or unbend. Transducers were attached to the rubber at the maximal bending position and fixed with a Tegaderm™ film from 3 M. Current signals were measured with a precise source measuring unit from Keysight (B2912A, 2 ch, 10 fA system).

**FEM simulation of the transversal-load test on elastic rubber**. The FEM analysis was performed with COMSOL Multiphysics. The stamp experiment shown in Fig. 3a was simulated for the case of rubber carrier using a purely structural mechanical model in two dimensions with a Cartesian coordinate system. As depicted in Supplementary Fig. 7a (inset), this system consists of a stiff flat stamp with rounded edge in contact with the UFPT (made up of a layer of P(VDF-TrFE) on top of the ultrathin parylene substrate with thickness $D_{sub}$) fixed onto the rubber substrate. A symmetry YZ-plane exists at $x = 0$ mm and the Z-depth was set to half the stamp side length $L_{sp}$. The top side of the stamp was vertically displaced, and the reaction force was multiplied by a factor of 4 to obtain the total stamp force $F$. The contact at the stamp-UFPT interface was modelled as an adhesive, reflecting the strong cohesion property of the UFPT. For the mechanical material properties, all layers were described as linear elastic and isotropic with the main parameters as Young modulus $Y$ and Poisson's ratio $v$. Supplementary Table 2 lists the chosen geometry and material values for all layers.

For P(VDF-TrFE), the mechanically induced piezoelectric polarisation in the poling direction (i.e., Y-direction in the material frame), $P_{piezo}$, was calculated following a dimensional model[69] as

$$P_{piezo}(x) = -\epsilon_{YY}(x) \cdot P_r, \qquad (7)$$

with $P_r$ being the remnant polarisation and $\epsilon_{YY}$ the local thickness-average of the simulated transversal strain in the piezoelectric layer upon deformation. The generated charge was calculated by integration along the transducer layer with side length $L_t$ according to

$$\Delta Q = -P_r \cdot L_{sp} \cdot 2 \int_0^{L_t/2} \epsilon_{YY}(x)\mathrm{d}x. \qquad (8)$$

Supplementary Fig. 7a shows the simulated volumetric strain (colour map) in the piezoelectric layer at a stamp displacement of $\Delta y = -50$ μm ($D_{sub} = 1$ μm), which corresponds to a total reaction force of 4.0 N. The derived sensitivity $S_{FEM} = 1.07$ nC N$^{-1}$. Outside the contact area, a tensile strain is dominant, which is caused by the severe deformation of the elastic rubber carrier. Below the stamp, a slight compression occurs, as expected. When decomposing the strain components in the piezoelectric layer into the transversal (i.e., out-of-plane) and longitudinal (i.e., in-plane) strain components $\epsilon_{YY}$ and $\epsilon_{XX}$, respectively, it becomes apparent that the main contribution to the piezoelectric response comes from the deformation in the outer vicinity of the stamp edge.

As to the FEM simulation of the transducer response and energy-conversion efficiency in bending mode B, a detailed description is given in Supplementary Note 4.

**OTFT-based rectifier circuits**. The structure of an OTFT-based diode is displayed in Fig. 6a. The metal electrodes are thermally evaporated through a shadow mask (60 nm of Al for the gate electrode and 50 nm of Au for the S/D electrodes). As a gate dielectric, a 10-nm-thick bilayer of alumina (formed by anodization[70]) and 12-dodecylphosphonic acid (TCI chemicals) were used. The SAM treatment was performed by immersing the sample in a 3-mMol 2-propanol solution of C$_{12}$-PA overnight. After this immersion step, the samples were carefully rinsed with pure solvent and blown dry by nitrogen. A 30-nm-thick dinaphtho[2,3-b:2,3-f]thieno [3,2-b]thiophene (DNTT) layer for the p-type OTFT-based diodes was deposited under high vacuum conditions on the gate dielectric layer. Before Au-electrode evaporation, holes were drilled with a green laser marker system (Keyence, T-Centric MD-T1000) to shorten the drain and gate contacts and to connect the organic diodes and capacitors to the energy-harvesting circuit.

**Electrical characterisation**. Organic diodes were electrically characterised using a semiconductor parameter analyser (Keysight, B1500A) and a SUESS probe station in a dark shield box. The thin-film capacitor measurements were performed by employing a chemical impedance analyser (HIOKI, IM3590, Agilent E4980). All electrical measurements were conducted under ambient conditions.

**Thin-film capacitor**. A 60–80-nm thick layer of aluminium was thermally evaporated through a shadow mask on the parylene substrate. The aluminium was then anodised to create a 10-nm-thick aluminium oxide layer (see Supplementary Fig. 18). Further, the top electrode was fabricated from thermally evaporated 60–80-nm-thick aluminium using shadow masking. The area of the thin-film capacitor was 0.5 cm². The LCR measurements revealed a capacitance of 0.23 μF ($f = 1$ kHz), which corresponds to a relative permittivity of $\varepsilon_{AlOx} \approx 9$ by assuming an alumina thickness of 10 nm.

**Pulse wave and pulse wave velocity (PWV) measurements**. A photograph of the measurement conditions can be found in Fig. 5. The ultraflexible transducer was conformally attached to the neck or the wrist of the arm and connected to a customised wireless measurement unit (24-bit analogue front end with up to 250-Hz sampling speed) with a resistor of 3.3 MΩ in parallel. To determine the PWV, two transducers were attached at a defined distance from one another on the neck, and the pulse wave signals were monitored with a Keysight (B2912A, 2 ch, 10 fA) system at a sampling rate of 5 kHz.

**Ultraflexible energy-harvesting device**. The energy harvesting system was fabricated by connecting one or a stack of piezoelectric nanogenerator(s) with a circuit comprising an organic rectifier and a thin-film capacitor (or conventional capacitor). An equivalent circuit diagram is shown in Supplementary Fig. 14. The connections between PENGs, rectifier circuits, and thin-film capacitors were realised with thin isolated metal wires (PTFE AWG26, 0.31 mm, KYOWA) and/or stretchable Ag-ink SX-ECA from CEMEDINE. It is important to note that the thin-film capacitors and rectifier circuits were fabricated on a single parylene (diX-SR) substrate.

**Human research participants**. To demonstrate the pulse wave monitoring by the ultrathin organic sensing device, one healthy female (age 32) and one healthy male (age 34) were tested. We have obtained informed consent from all participants. All experiments regarding pulse wave monitoring complied with guidelines by Osaka University Research Ethics Committee.

## Data availability
The authors declare that the data supporting the findings of this study are available within the article and its Supplementary Information files. Additional data are available from the corresponding authors upon reasonable request.

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

## Acknowledgements
This work has been supported financially by the Austrian Science Fund (FWF) (J4145-N30), the Center of Innovation program of the Japan Science and Technology Agency (JST), the Japan Society for the Promotion of Science (JSPS) KAKENHI and by New Energy and Industrial Technology Development Organization (NEDO). A part of this work was supported by Nanotechnology Platform of MEXT, Grant Number JPMXP09S20OS0019. We would like to specially thank Maria Belegratis (Joanneum Research) for preparing the P(VDF:TrFE) solutions and Roland Resel (TU Graz) for discussing the XRD results. We also acknowledge Cemedine Corporation for the provision on stretchable Ag-ink.

## Author contributions
A.P., B.S., E.K.-P., P.S., T.U., T.A. and T.S. conceived the concept, processing and structure details. A.P., E.K.-P., T.U. and T.A. carried out the device development, experimental work and analysed the data. P.S. carried out the FEM simulations and wrote this part of the paper. A.P., B.S., P.S., E.K.-P., T.U., T.A. and T.S. wrote the paper. All authors commented on the paper.

## Competing interests
The authors declare no competing interests.
