## [Peer Review file · Nature Communications]

Reviewer #1 (Remarks to the Author):

This is a very interesting paper on high performance piezoelectric energy harvesting devices integrated on ultraflexible substrates that can be mounted onto human skin. The experimental work is carefully performed and I recommend that the paper is published after the following minor comments are taken into consideration:

- I find it difficult to assess from the information given in the paper whether the proposed energy harvesting device could deliver sufficient power for a realistic application, such as measuring blood pressure or heartrate as suggested in the paper. Would the mechanical motion naturally present on the skin be sufficient to harvest enough energy to power a moderately complex electronic circuit ?
- In this respect it might be helpful to quote the performance of the energy harvester not only as a volumetric power density, but also as an areal power density using the actual thickness of the device. It might also be helpful to estimate/calculate the energy efficiency for conversion of the mechanical energy provided by the external force into electrical energy stored in the capacitor.
- As explained on page 14 the charge response of the transducer on flexible substrates is determined by effects at the edge of the stamp. What does this imply for the optimum mode of operation when the device is mounted on human skin in a realistic application ?
- Estimating the degree of crystallinity of semicrystalline polymers is notoriously difficult and the method used by the authors is likely to have a large uncertainty. I found Ref. 54 not very helpful to justify the approach and I would recommend providing a more careful justification of the method used for determining the degree of crystallinity.

Reviewer #2 (Remarks to the Author):

In this work, the authors Petritz et al. have developed ultra-thin plastic mechanical sensors based on piezo-electric transducers. In addition, they have integrated organic diodes as rectifying elements together in an ultra-thin form factor. The monolithic integration work is exciting, and is novel in wearable mechanical sensors and energy generators. I believe that this work is of the right quality for this journal. However, I have some questions that needs to be addressed before publication.

1. In Fig 3b, there seems to be some baseline fluctuations in the charge output in the time-series study. Can the authors explain these fluctuations?
2. What is the temperature sensitivity of the device relevant to physiologic parameters?
3. In terms of the size of the piezoelectric transducers, where would the scaling limit be for this particular readout circuit? Going to smaller areas would likely negatively impact the sensitivity, and it would be helpful to know what the limits are.
4. It would be good for the authors to compare this new device and PWV obtained with other types of sensors. e.g. recent work in Proc. Natl. Acad. Sci. 202010989 (2020).
doi:10.1073/pnas.2010989117
5. In terms of the variance in the performance of the organic devices, it would be helpful to give a histogram of the rectifying performance across a few devices.

Author's reply to Reviewers

First, we would like to thank all reviewers for their invaluable and constructive comments. Because all comments are very important, we have revised the manuscript in accordance with these comments. The revised portions are highlighted in "Red text" in the manuscript (Main text and Supplementary Information (SI)).

Response to Reviewer's comments

Reviewer #1:

This is a very interesting paper on high performance piezoelectric energy harvesting devices integrated on ultraflexible substrates that can be mounted onto human skin. The experimental work is carefully performed and I recommend that the paper is published after the following minor comments are taken into consideration:

Comment (#1-1): *I find it difficult to assess from the information given in the paper whether the proposed energy harvesting device could deliver sufficient power for a realistic application, such as measuring blood pressure or heartrate as suggested in the paper. Would the mechanical motion naturally present on the skin be sufficient to harvest enough energy to power a moderately complex electronic circuit?*

Reply to comment (#1-1): We would like to thank Reviewer #1 for their positive evaluation. We sincerely welcome the important comment. The text in the manuscript is improved to read:

Main text, page 26, line 530: The paragraph "From the energy levels generated during bending motions (mode B), we roughly estimated that more than 200 mJ per day can be gained from biomechanical motions if multi-layer UFPTs are placed on joints like knees or elbows. For a detailed description please be referred to the Supplementary Information in Chapter 18." is added.

Main text, page 30, line 599: The paragraph "With regard to power supply for e-health patches, we estimate that energy levels of more than 200 mJ per day can be reached by energy harnessing from biomechanical motions if multi-layer UFPTs are placed on joints like knees or elbows. This is sufficient to power a wireless electronic system operating in an ultra-low power consuming duty-cycled fashion⁶⁸ and should allow to transmit the measured pulse wave data several times a day (e.g. once or twice an hour). Yet, these values are based on several assumptions, thus further research and extended field tests are necessary to test the long-term energy harvesting potential of our technology on different parts of the body." is added.

References added:

68. Moreno-Cruz, F. *et al.* treNch: Ultra-Low power wireless communication protocol for IoT and energy harvesting. *Sensors* **20**, 6156 (2020).

Supplementary Information, page 22, line 304: The section "**19. Estimation of energy harnessing performance from biomechanical motion of multi-layer UFPTs:** The energy delivered by our piezoelectric energy harvesting device strongly depends on the mounting position on the human body. When actuated by bending, it is important that a high stretching is introduced to the transducer. Therefore, places subject to a lot of muscle work, such as when the leg or arm muscles expand and contract, or where strong bending appears, like at the elbow joint or knee joint, could be ideal mounting spots for maximizing the harvesting. Another good position could be on/under the sole, which deforms a lot when walking. However, we admit that further experiments are necessary to investigate the UFPT harvesting performance on the different positions on the human body.

The amount of energy harvested from the motion of the elbow joint can be roughly estimated from our bending tests on a rail, as shown in Supplementary Figure S15. Here the deformation of the rubber during bend and unbend positions is somehow similar to the elbow joint movement between flexion and extension position. The mean generated areal energy density E_{gen} for a single transducer layer was measured to be around 20 nJ cm^{-2} per cycle (bend and release). This was calculated by $E_{\text{gen}} = E / (N \cdot A)$ where E is the stored energy in the capacitor (from Fig. 7d, $C = 10 \text{ }\mu\text{F}$), N the number of bending cycles (450) and A (2.25 cm^2) the transducer area.

Stacking the ultraflexible transducers would be a promising approach for increasing the energy output as we demonstrated in this work. The generated charge level was either doubled for the two-layer stack or even tripled for the three-layer stack as compared to the maximum charge level measured for a single-layer UFPT under the same excitation conditions (Fig. 3d).

Placing a multiple layer stack of 25 transducers (yet amounting to a just $60 \text{ }\mu\text{m}$ thin device) with an active area of 20 cm^2 on the elbow joint would generate about $10 \text{ }\mu\text{J}$ of energy per motion cycle (movement between flexion and extension position). Thus, for 100 movements / hour we estimate a harvested energy of 1 mJ per hour or 16 mJ per day if we assume an activity period of 16 hours (for storage capacitors with too small capacitance discharging effects might decrease these values to a certain extent).

Another good position to place the UFPT harvester would be the knee joint. Per day, an average person takes 2000-4000 steps in normal activities and over 10000 steps in sporting activities; walking would allow harvesting 20 to 100 mJ per day for one knee.

We believe that a feasible application scenario would be to continuously harvest and store biomechanical energy in an energy storage device until a certain charging level is reached. Then a pulse wave measurement will be triggered and the measured data will be stored in a data logger. Although in our opinion it is unlikely to achieve a continuous, uninterrupted recording of vital parameters solely based on harvested energy, the charging period with a multiple stack of UFPTs can still be kept quite short (a few minutes), so as to allow for a periodic health tracking. The threshold charging level can be further adjusted to enable a wireless transfer of data to a computer or smartphone once or twice a day.

The update period critically depends on the power consumption of the electronic circuit. For this application, we would need a compact wireless electronic system operating in a special

duty-cycle that allows ultra-low power consumption by remaining in a low/zero energy consumption state (sleep phase) most of the time and just consuming energy during measurements and communication in the active phase. We do not develop such a low power system but there are examples already presented in literature that can be suitable for this purpose. Ultra-low power wireless communication protocols with outstanding consumption figures of less than 300 nW and 1 mJ for the sleep and active phases, respectively were reported^{S11}. A comprehensive overview of different energy harvesting strategies is published in Singh et al.^{S12}.

To summarize, from biomechanical motions we estimate that an energy amount of up to 200 mJ per day can be harvested, which may be sufficient to power a wireless electronic system in a duty-cycled operation and enables to transmit the measured pulse wave data several times a day (every 30-60 mins). These values are based on some partly rough assumptions, thus further research is necessary to investigate the real harvesting performance of UFPT's on human skin (maybe not only joints are good placing spots, muscle work (expansion and contraction) may also generate high signals...). In addition, a suitable wireless data processing and communication system has to be developed." is added.

Comment (#1-2a): *In this respect it might be helpful to quote the performance of the energy harvester not only as a volumetric power density, but also as an areal power density using the actual thickness of the device.*

Reply to comment (#1-2a): Yes, we agree with the reviewer's comment. In the answer of the previous question, we considered the areal instead of the volumetric power density to estimate the generated energy for a realistic application. We also added the areal power density for the three different bendings modes. The manuscript is improved to read:

Main text, page 25, lines 505-507: the sentence "By varying the external resistance, the maximum output power reached $P_{out,max} \sim 3.2 \text{ mW cm}^{-3}$ (related to the total PENG volume) for an optimum load..." is changed to "By varying the external resistance, the maximum volumetric output power density reached $P_{out,max} \sim 3.2 \text{ mW cm}^{-3}$ (corresponding to an areal power density of $P_{out,max} \sim 0.8 \text{ } \mu\text{W cm}^{-2}$) for an optimum load...";

Main text, page 26, line 518: areal power density "(0.25 $\mu\text{W cm}^{-2}$)" is added;

Main text, page 26, line 532: areal power density "(0.3 $\mu\text{W cm}^{-2}$)" is added;

Main text, page 30, line 595: areal power density "(0.75 $\mu\text{W cm}^{-2}$)" is added;

Comment (#1-2b): It might also be helpful to estimate/calculate the energy efficiency for conversion of the mechanical energy provided by the external force into electrical energy stored in the capacitor.

Reply to comment (#1-2b): We appreciate the valuable suggestion to estimate the energy efficiency of our harvesting device. In similar previous studies (cf. Dagdeviren et al., *Extrem. Mech. Lett.* 2016, 9, 269., and references therein)¹, the efficiency was calculated as the ratio of electrical energy stored in the harvester (W^{el}) to the total mechanical input energy (W^m), considering also the work performed on the underlying layer causing deformation of the harvester. However, the invested work to deform the underlying layer depends strongly on the layer's mechanical properties, dimension, deformation shape etc. and varies among possible use cases. Therefore, we find it more appropriate to evaluate the ratio of output electric energy to the mechanical work performed on the UFPT, consisting of the piezoelectric layer and the substrate. The overall efficiency can then be written as

$$\eta = \eta^{el} \cdot \eta^m \quad (1)$$

with

$$\eta^m = \frac{W_t^m}{W_{sub}^m + W_t^m} \quad (2)$$

and

$$\eta^{el} = \frac{W_t^{el}}{W_t^m} \quad (3)$$

Here, W_t^{el} and W_t^m denote the stored electrical energy and mechanical strain energy in the piezoelectric material, respectively, and W_{sub}^m is the mechanical strain energy stored in the passive substrate.

We both performed 3D FEM simulation of the bending actuation use case (mode B) as well as derived an analytical model to make an estimation of η^m , η^{el} and also the overall energy conversion efficiency. The validity of the FEM simulation was first tested by comparison of calculated and experimental electrical energy amounts per bending cycle. While the peak power density derived by FEM was almost the same as in the experiment (1.1 mW/cm³ vs. 1.0 mW/cm³), the calculated mean energy amount/cycle was 3.6 times higher than in the experiment. We assume that some deviations from the actual load profile and adhesion condition during the experiment, which are difficult to be exactly reproduced in the FEM, are the main reasons for the deviation. Given the quite fair agreement and the fact that the simulation was performed under quasistatic conditions to derive the energy quantities, we find that the use of the FEM simulation for estimation of the energy quantities is justified.

For the dimension of our presented UFPT, the estimated energy conversion efficiency according to equation (1) amounted to **0.139 %** according to the simulation, while the analytical model gives an upper limit of 0.185%. We further investigated the influence of substrate thickness on the energy efficiency. In that case, the values predicted by the analytical model, especially for the mechanical

¹ Dagdeviren, C. *et al.* Recent progress in flexible and stretchable piezoelectric devices for mechanical energy harvesting , sensing and actuation. *Extrem. Mech. Lett.* **9**, 269–281 (2016).

energy ratio η^m , get very close to those derived from the more complex and accurate 3D simulation. According to the model, the ratio η^m can be estimated as

$$\eta_{th}^m = \frac{W_t^m}{W_{sub}^m + W_t^m} = \frac{r_{th}}{(1+r_{th})} \quad (4)$$

with

$$r_{th} = \frac{W_t^m}{W_{sub}^m} = \frac{D_t}{D_{sub}} \cdot \frac{E_t}{E_{sub}} \cdot \frac{1-\nu_{sub}^2}{1-\nu_t^2} \quad (5)$$

where E_t , ν_t and E_{sub} , ν_{sub} are the Young's modulus and Poisson's ratio of the piezoelectric and the substrate layer, respectively, and D_t , D_{sub} the respective layer thicknesses. The main advantage of a thin (and soft) substrate is the reduced energy loss during deformation. As to the energy conversion efficiency of the piezoelectric layer itself, η^{el} , we found the following approximation for P(VDF-TrFE) under uniaxial bending with clamping condition in the orthogonal direction:

$$\eta_{th}^{el} = \frac{P_r^2}{\epsilon_0 \epsilon_r - \frac{\nu_t^2 P_r^2}{E_t}} \cdot \frac{\nu_t^2 (1+\nu_t)}{E_t (1-\nu_t)} \quad (6)$$

with P_r being the remnant polarization and ϵ_r the relative permittivity of the poled layer. For our UFTP it amounts to $\eta_{th}^{el} = 0.36\%$.

The main summary of the theoretical findings and calculation of the energy conversion efficiency were added to the main text:

Main text, page 27, line 540: The section “A vital parameter in terms of energy harvesting is the efficiency of conversion from mechanical to electrical energy²³. The total mechanical input energy needed to cause an actuation of the harvester can hardly be determined as it strongly depends on the body the harvester is attached to in terms of its mechanical properties, dimension, deformation shape and other quantities. However, one can define the energy conversion efficiency as the ratio of harvested electrical energy to the stored strain energy upon deformation of the whole harvester system, including the passive substrate, i.e.

$$\eta = \frac{W_t^{el}}{W_t^m} \cdot \frac{W_t^m}{W_{sub}^m + W_t^m} \quad (3)$$

where W_t^{el} / W_t^m is the conversion ratio of mechanical strain energy (W_t^m) to electrical energy (W_t^{el}) given by the piezoelectric material, and W_{sub}^m is the mechanical energy stored in the passive substrate. We performed 3D FEM simulations of the bending experiment (mode B) to estimate the energy levels for different substrate thicknesses in this actuation mode and compared the results with an analytical model (see Chapter 19 in the Supplementary Information). Obviously, a large strain energy ratio W_t^m / W_{sub}^m is essential for a high conversion efficiency, which can be approximated by

$$\frac{W_t^m}{W_{sub}^m} \approx \frac{D_t}{D_{sub}} \cdot \frac{E_t}{E_{sub}} \cdot \frac{1-\nu_{sub}^2}{1-\nu_t^2} \quad (4)$$

with E_t , ν_t and E_{sub} , ν_{sub} being the Young's modulus and Poisson's ratio of the piezoelectric and the substrate layer, respectively, and D_t , D_{sub} denoting the respective layer thicknesses (cf. Fig. S20a). The energy ratio thus scales inverse with the product $D_{sub} \cdot E_{sub}$. For the

presented UFTP in bending actuation, the model given by Equ. (3) and (4) predicts a conversion efficiency of $\eta = 0.185 \%$, whereas the more accurate 3D simulation gives $\eta = 0.139 \%$. From Fig. S20b it is clear that a very small substrate layer thickness is significant to obtain a high energy conversion efficiency. When using a 10 times thicker substrate, i.e. $10 \mu\text{m}$, the simulated efficiency significantly drops to only 0.018% , which is more than 7 times lower, and for a $100 \mu\text{m}$ thick substrate it is even 25 times lower. This highlights the major improvement in energy conversion of P(VDF-TrFE)-based transducers by drastically decreasing their substrate thickness and thus strongly supports the concept of ultraflexibility.” is added.

Details to the FEM simulation of the bending actuation with results as well as the derivation of the analytical model were added to the amended Supplementary Information under Chapter 19.

Supplementary Information, page 22, line 305: The chapter “**19. Calculation of the UFPT’s energy conversion efficiency:** In similar previous studies (cf. Dagdevieren et al. and references therein)¹⁰, the energy conversion efficiency of a transducer was calculated as the ratio of energy stored in the harvester to the total mechanical input energy, considering also the work performed on the underlying layer causing deformation of the harvester. However, the invested work to deform the underlying layer depends strongly on the layer’s mechanical properties, dimension, deformation shape etc. and varies among possible use cases. Therefore, we find it more appropriate to evaluate the ratio of output electric energy to the mechanical work performed on the UFPT, consisting of the piezoelectric layer and the substrate. The overall efficiency can then be written as

$$\eta = \eta^{el} \cdot \eta^m \quad (\text{S1})$$

with

$$\eta^m = \frac{W_t^m}{W_{sub}^m + W_t^m} \quad (\text{S2})$$

and

$$\eta^{el} = \frac{W_t^{el}}{W_t^m} \quad (\text{S3})$$

Here, W_t^{el} and W_t^m denote the stored electrical energy and mechanical strain energy in the piezoelectric material, respectively, and W_{sub}^m is the mechanical strain energy stored in the passive substrate.

Since the stored strain energy during deformation is not accessible through experiment, we applied a three-dimensional FEM simulation to *numerically derive the respective energy quantities* for the case of bending on a rubber layer (mode B) and also compared them with the predictions of an *analytical model*.

FEM simulation: The FEM model of the transducer was the same as for the transversal load simulation, see Methods, Chapter 7 and Table S2. The relative permittivity, ϵ_r , of the piezoelectric layer was taken to be 8.5. The transducer was placed centrally on top of a $10 \times 10 \text{ cm}^2$ rubber sheet with 2 mm thickness (with a Young’s modulus of 1.45 MPa and a Poisson ratio of 0.49) without allowing for friction or sliding. Clamping conditions were applied on

both ends of the rubber over an area of $2 \text{ cm} \times 1 \text{ cm}$ mimicking the metal clamps in the experiment. Symmetry was employed in the xz - and yz -planes normal to the rubber surface to reduce the computational complexity by a factor of ≈ 4 . The lateral displacement of the clamped region causing upward displacement and convex bending of the rubber was simulated with displacement values used during the experiment. Figure S19a depicts the model at maximum bending (cf. Figure S16a, photograph at maximum bending). To calculate the output currents and electric energy, the reduced transducer element was virtually either short-circuited or connected to a load resistance of $R_{L, \text{sym}} = 4 \cdot 2.5 \text{ M}\Omega$, which corresponds to the optimum load under experimental condition of $2.5 \text{ M}\Omega$ (cf. Figure S16b), where the factor 4 is due to the symmetry of the model. A time study step was performed with a triangular displacement profile over a period of 0.5 s (onset at $t = 0.25 \text{ s}$), which corresponds to the average excitation frequency in the experiment (2 Hz). The time plot in Figure S19b shows the currents at short-circuit and load condition, respectively, as well as the power dissipated by the load normalized by the transducer volume for direct comparison with the experimental values in Figure S16b. The volumetric peak power density amounts to $\approx 1.1 \text{ mW cm}^{-3}$ and is in excellent agreement with the experimental value (cf. Table 1). The electrical energy per area for a single load cycle was calculated by a time integration of the power dissipated over the load resistance and amounted to 72 nJ cm^{-2} , which is 3.6 times higher than in the experiment. This discrepancy in total energy output per cycle might be due to a different load profile in the experiment or non-perfect adhesion of the UFPT on the substrate in the experiment causing perhaps a transducer displacement or slipping in the initial actuation cycle, which is difficult to account for in the simulation. The performed mechanical work was numerically calculated for the piezoelectric and substrate layer, respectively, as (Einstein summation convention applied)

$$W^m = \iint \sigma_i d\epsilon_i dV \quad (\text{S4})$$

Static study steps were performed with varying substrate thicknesses D_{sub} to derive a trend of the energy conversion coefficients η^m and η^{el} at full bending of the rubber layer. The internally stored electric energy at full bending, $W^{el} = \frac{\Delta Q_t}{2 C_t}$, with displacement charge ΔQ_t and the transducer's capacitance C_t , was used to derive η^{el} in the static case.

Supplementary Figure S19 | FEM simulation of the bending actuation (mode B) a) 3D representation of the model at full bending with stress levels in color. b) Time evolution of short circuit current ($I_{s.c.}$), load current (I_{load}) and calculated output power density for $R_L = 2.5 \text{ M}\Omega$.

Analytical model: To derive the *theoretical energy conversion efficiency of the piezoelectric P(VDF-TrFE), η_{th}^{el}* , we consider the case of uniaxial in-plane stretching with clamping applied to the lateral direction. This shall mimic the situation of our UFPT being adhered to a deforming surface (in the xy -plane) and undergoing a unidirectional stretching (x -direction) due to bending of the surface (bending axis pointing in y -direction). The piezoelectric constitutive equations in pseudovector form are^{S13}

$$\epsilon_p = s_{pq}^E \sigma_q + d_{ip} E_i \quad (S5)$$

$$D_i = d_{iq} \sigma_q + \epsilon E_i \quad (S6)$$

where ϵ and σ are the strain and stress pseudovectors, s^E is the compliance matrix at constant field, $\epsilon = \epsilon_0 \epsilon_r$ is the permittivity, D the electric displacement, E the electric field and d the matrix with piezoelectric coefficients ($i=1, 2, 3$ and $p, q = 1...6$). For the case of uniaxial loading in the x - or 1-direction, the clamping condition leads to the following boundary conditions: $\epsilon_2 = 0, \sigma_3 = 0$. In addition, no shear strains/stresses shall appear. The polarization points to the 3-direction. With electrodes applied on the top and bottom, only $E_3 \neq 0$. Applied to Equ. (S5), we obtain:

$$\epsilon_1 = s_{11}^E \sigma_1 + s_{12}^E \sigma_2 + d_{31} E_3 \quad (S7)$$

$$\epsilon_2 = s_{21}^E \sigma_1 + s_{22}^E \sigma_2 + d_{32} E_3 = 0 \quad (S8)$$

The performed *mechanical work* per volume during deformation of the piezoelectric layer, w^m , is (where volumetric changes due to bending are neglected)

$$w^m = \int \sigma_1 d\epsilon_1 \quad (S9)$$

Applying Equ. (S7) and (S8) for constant voltage condition ($dE_3 = 0$) yields

$$w^m = \left(s_{11}^E - \frac{s_{12}^E s_{12}^E}{s_{22}^E} \right) \cdot \frac{\Delta \sigma_1^2}{2} \quad (S10)$$

The electrical field generated during deformation can be obtained from (S6) at open-circuit condition ($D_3 = 0$) and amounts to

$$E_3 = - \frac{\left(d_{31} - d_{32} \frac{s_{12}^E}{s_{22}^E} \right)}{\left(\epsilon - \frac{d_{32}^2}{s_{22}^E} \right)} \cdot \Delta \sigma_1 \quad (S11)$$

When connected to a perfect load, the available *electrical energy* per volume, w^{el} , during release is then

$$w^{el} = \int E_3 dD_3 = \int_{\Delta \sigma_1}^0 E_3 \left(d_{31} - d_{32} \frac{s_{12}^E}{s_{22}^E} \right) d\sigma_1 = \frac{\left(d_{31} - d_{32} \frac{s_{12}^E}{s_{22}^E} \right)^2}{\left(\epsilon - \frac{d_{32}^2}{s_{22}^E} \right)} \cdot \frac{\Delta \sigma_1^2}{2} \quad (S12)$$

Thus, for the energy conversion ratio we get:

$$\eta_{th}^{el} = \frac{w^{el}}{w^m} = \frac{\left(d_{31} - d_{32} \frac{E}{s_{22}^E}\right)^2}{\left(\epsilon - \frac{d_{32}^2}{s_{22}^E}\right) \left(s_{11}^E - \frac{s_{12}^2}{s_{22}^E}\right)} \quad (S13)$$

Using the isotropic model for P(VDF-TrFE) (see Chapter 7) and applying the dimension model for its piezoelectricity (with piezoconstants $e_{33} = -P_r$, $e_{31} = e_{32} = 0$)^{S14}, the piezoelectric coefficients d_{ij} can be obtained from the compliance matrix and the remnant polarization P_r as

$$d_{3j} = -P_r s_{3j}^E \quad (S14)$$

With this, Equ. (S13) becomes

$$\eta_{th}^{el} = \frac{P_r^2}{\epsilon_0 \epsilon_r - \frac{v_t^2 P_r^2}{E_t}} \cdot \frac{v_t^2 (1+v_t)}{E_t (1-v_t)} \quad (S15)$$

where E_t is the Young's modulus and v_t is the Poisson ratio of the P(VDF-TrFE).

For the sample used at the bending test (see Table S2) this gives $\eta_{th}^{el} = 0.36\%$, corresponding to a mechanical coupling coefficient $k_{31}^w = \sqrt{\eta_{th}^{el}} = 0.06$ ^{S13}.

The *mechanical energy efficiency*, which we define here as $\eta^m = \frac{W_t^m}{W_{sub}^m + W_t^m}$, (cf. Equ. (3)) can be derived as follows. Using Equ. (S9), the total mechanical work performed during (elastic) bending of a layer at radial position $z = z_l$, thickness D , length l , and width b is

$$W^m = \int w^m dV = b l \int_{z_l}^{z_l+t} \sigma_1 d\epsilon_1 dz \quad (S16)$$

The strain introduced by bending at a radius R is $\epsilon_1(z) = (z - z_N)/R$, where z_N is the radial z-position of the neutral mechanical plane (NMP). Using $\sigma_1 = Y_{eff} \epsilon_1$ we obtain

$$W^m = \frac{b l}{2} t E_{eff} \frac{d^2}{R^2} \left(1 + \frac{t}{d} + \frac{t^2}{3d^2}\right) \quad (S17)$$

with $d = z_l - z_N$ being the radial distance to the NMP. The effective Young's modulus, E_{eff} , depends again on the clamping condition and material property. For an isotropic material and with the same clamping condition as above ($\epsilon_2 = 0$), one gets $E_{eff} = E/(1 - v^2)$. In the case of $t \ll d$, Equ. (S17) reduces to

$$W^m = \frac{b l}{2} \frac{t E}{1-v^2} \frac{d^2}{R^2} \quad (S18)$$

Next, we can derive Equ. (4) as the ratio of mechanical work stored in the piezoelectric layer (thickness D_t) and the substrate layer (thickness D_{sub}) assuming both have the same footprint (i.e. $l = L_t = L_{sub}$, $b = b_t = b_{sub}$):

$$r_{th} = \frac{W_t^m}{W_{sub}^m} = \frac{D_t}{D_{sub}} \cdot \frac{E_t}{E_{sub}} \cdot \frac{1-v_{sub}^2}{1-v_t^2} \quad (4)$$

The theoretical mechanical energy efficiency is then

$$\eta_{th}^m = \frac{r_{th}}{(1+r_{th})} \quad (S19)$$

Obviously, it scales with the ratio of piezoelectric layer thickness vs. substrate thickness, D_t/D_{sub} . As can be seen in Figure S20a, the mechanical energy efficiency as predicted by the model is also in excellent agreement with the numerically calculated values based on the 3D FEM simulation. For the presented UFPT with an only 1 μm thin substrate, it amounts to 52 %. The overall theoretical energy efficiency for the presented UFPT is $\eta_{th} = \eta_{th}^{el} \cdot \eta_{th}^m = 0.185$ %. From the FEM we obtain $\eta = 0.139$ %. The theoretical values are slightly larger compared to the values calculated with FEM (Figure S20b). In the FEM simulation the clamping conditions and bending of the transducer are closer to the real situation and thus more complex, which obviously reduces the effective electromechanical coupling in the piezoelectric layer (η^{el}).

Supplementary Figure S20 | Energy conversion efficiency of the UFPT (Mode B). Mechanical energy efficiency (a) and overall energy conversion efficiency (b) derived from the 3D FEM simulation and comparison with the analytical model for varying substrate thicknesses D_{sub} .'' is added.

Comment (#1-3): *As explained on page 14 the charge response of the transducer on flexible substrates is determined by effects at the edge of the stamp. What does this imply for the optimum mode of operation when the device is mounted on human skin in a realistic application?*

Reply to comment (#1-3): Thank you for the important question. The main finding of our transversal load experiments was that a softer elastic carrier substrate induces more stress/strain in the piezoelectric film near the edge of the stamp and results in a strongly enhanced response signal for a given transversal load in contrast to a rigid substrate. This in turn will significantly increase the output signal from the transducers. It should be noted, however, that this observations are especially true for vertical load like pressing on the transducer. In a real application scenario for energy harvesting from biomechanical excitations the edge effect is not that relevant as here the strain is mainly induced by bending loads and not that much by transversal loads. This transversal load actuation were performed for fundamental sensor characterization (sensitivity,...) and not meant to simulate biomechanical excitations.

Comment (#1-4): *Estimating the degree of crystallinity of semicrystalline polymers is notoriously difficult and the method used by the authors is likely to have a large uncertainty. I found Ref. 54 not very helpful to justify the approach and I would recommend providing a more careful justification of the method used for determining the degree of crystallinity.*

Reply to comment (#1-4): We sincerely welcome the reviewer comment and we agree. In order to better justify this approach the old Ref. 54 was removed, two new references were added, and the text is improved to read:

Main text, page 10, line 194: Ref. 54 was removed and two new reference are added;

52. Mahdi, R., Gan, W. & Majid, W. Hot Plate Annealing at a Low Temperature of a Thin Ferroelectric P(VDF-TrFE) Film with an Improved Crystalline Structure for Sensors and Actuators. *Sensors* **14**, 19115–19127 (2014).
54. Kavesh, S. & Schultz, J. M. Meaning and measurement of crystallinity in polymers: A Review. *Polym. Eng. Sci.* **9**, 452–460 (1969).

Main text, page 10, line 200: the sentence “This approach provides an estimation of the crystallinity in a polymer and is referred to as “apparent” crystallinity in literature⁵⁴.” is added;

Reviewer #2:

In this work, the authors Petritz et al. have developed ultra-thin plastic mechanical sensors based on piezo-electric transducers. In addition, they have integrated organic diodes as rectifying elements together in an ultra-thin form factor. The monolithic integration work is exciting, and is novel in wearable mechanical sensors and energy generators. I believe that this work is of the right quality for this journal. However, I have some questions that needs to be addressed before publication.

Comment (#2-1): *In Fig 3b, there seems to be some baseline fluctuations in the charge output in the time-series study. Can the authors explain these fluctuations?*

Reply to comment (#2-1): We would like to thank Reviewer #2 for their positive evaluation and the very important comments. The baseline fluctuations of the charge output may have several reasons. One reason for the baseline drift could be the temperature sensitivity (pyroelectricity) of our ferroelectric transducers. For more details and an explanation to overcome this issue, please see the answer to question two.

Another origin for baseline fluctuations can be electrostatic coupling of moving charged objects (vibration) near our test setup. To reduce this effect, we electrically shielded our measurement setup und used low noise cables. Another optimization, which we installed recently, was a Faraday cage around the tested sample in our tensile tester setup. It is also important to mention here that we cross-checked whether the measured charge stems from the piezoelectric effect or involves a contribution from some electrostatic coupling as well (triboelectric effect). Therefore, we compared measurements from non-poled and poled samples and found that the charge response for the non-poled transducer samples was negligible (< 1 pC/N).

Accordingly, we attribute the baseline fluctuations to charge generation stemming from thermal fluctuations. The text in the manuscript is improved to read:

Main text, page 14, line 263: The sentence “**The small baseline fluctuation is stemming from charges generated by thermal fluctuations.**” is added;

Comment (#2-2): *What is the temperature sensitivity of the device relevant to physiologic parameters?*

Reply to comment (#2-2): Thank you for the important point. We sincerely welcome the reviewer's comment. Due to the pyroelectricity of the UFPTs, which is an intrinsic property of any ferroelectric material, also temperature changes/fluctuations ΔT can be registered with a sensitivity of about - 42 to - 48 $\mu\text{C m}^{-2} \text{K}^{-1}$.² Thus, it is also possible to measure changes of the body-temperature with our ferroelectric transducer. However, it is not possible to measure absolute temperature values. We are now working on a specific dual-gate ferroelectric sensor architecture technology, which allows a bimodal sensing of temperature and pressure (static and dynamic). With this device, a simultaneous measurement of the human pulse wave and the body temperature will be feasible.

The strong ΔT -sensitivity may rise the question of crosstalk between the thermal and pressure sensing modes. In our application, we might expect a distortion of the pulse wave measurement by the unavoidable pyroelectric effect, which we did not observe as demonstrated in Supplementary Video S1, where a stable measurement of the human pulse wave is shown. Thus, the influence of the pyroelectric effect occurs to be negligible for this application. It should also be noted that fluctuations of the body temperature appear at much lower frequencies compared to pulse waves and can thus be easily discriminated by a smart data evaluation routine, e.g. based on machine learning. In order to completely avoid the cross-sensitivity, a ferroelectric nanocomposite layer made of inorganic ferroelectric nanoparticles like PbTiO_3 or sodium bismuth titanate ($\text{NaBiTi}_2\text{O}_6$ or BNT) dispersed in a P(VDF-TrFE) matrix can be used as an alternative to the PVDF-TrFE copolymer, where either the piezo- or the pyroelectric effect is suppressed by selective and material tailored poling procedures.³

² Stadlober, B., Zirkel, M. & Irimia-Vladu, M. Route towards sustainable smart sensors: ferroelectric polyvinylidene fluoride-based materials and their integration in flexible electronics. *Chem. Soc. Rev.* **48**, 1787–1825 (2019).

³ Ali, T. A. *et al.* Screen-Printed Ferroelectric P(VDF-TrFE)-co-PbTiO₃ and P(VDF-TrFE)-co-NaBiTi₂O₆ Nanocomposites for Selective Temperature and Pressure Sensing. *ACS Appl. Mater. Interfaces* **12**, 38614–38625 (2020).

Comment (#2-3): In terms of the size of the piezoelectric transducers, where would the scaling limit be for this particular readout circuit? Going to smaller areas would likely negatively impact the sensitivity, and it would be helpful to know what the limits are.

Reply to comment (#2-3): Thank you for the important comment. We also measured the human pulse wave with transducers with much smaller active sensing areas of only 3.24 mm^2 ($1.8 \text{ mm} \times 1.8 \text{ mm}$). In Fig. S17b the human pulse wave measurements on the wrist from transducers with two different active sensing areas are compared, showing no difference in sensitivity.

Ultraflexible imperceptible sensors with an active area of only a few mm^2 have excellent sensitivity for bio-signal monitoring. However, by further decreasing the sensors to much smaller sizes, less excited sensing area from the pulse wave will result in a strongly reduced sensor response. Ultraflexible organic amplifiers can be an option for processing weak physiological signals with high signal integrity and sensitivity. The ultraflexible organic amplifier can be easily combined with the sensors (single substrate integration) and can be conformably attached to the human body, enabling a direct amplification near the signal edge as previously reported^{4,5}. A Figure is added in the Supplementary Information and the text in the manuscript is improved to read:

Main text, Page 18, line 375: The sentence “Additionally, we used this setup to monitor the pulse wave on the wrist, whereby a pulse rate of 60 min^{-1} was extracted (see Fig. S11).” is added;

Supplementary Information, Page 14, line 171: The chapter “10. Pulse wave measurements on the wrist” with “Figure S11” is added;

Supplementary Figure S11 | Pulse wave measurements on the wrist. (a) Photograph of pulse wave measurement with an ultra-flexible piezoelectric transducer conformable attached on the wrist and connected to a wireless module with a $3.3 \text{ M}\Omega$ resistor parallel. (b) The human pulse wave associated with the flow of blood through near-surface arteries was monitored by a patch with an active sensing area of 2.25 cm^2 (left) and 3.24 mm^2 (right), respectively. A pulse rate of 60 min^{-1} could be extracted for the large sensor area and 63 min^{-1} for the small sensor area.

⁴ Sekitani, T. et al. Ultraflexible organic amplifier with biocompatible gel electrodes. *Nat. Commun.* **7**, 11425 (2016).

⁵ Sugiyama, M. et al. An ultraflexible organic differential amplifier for recording electrocardiograms. *Nat. Electron.* **2**, 351–360 (2019).

Comment (#2-4): *It would be good for the authors to compare this new device and PWV obtained with other types of sensors. e.g. recent work in Proc. Natl. Acad. Sci. 202010989 (2020). doi:10.1073/pnas.2010989117*

Reply to comment (#2-4): We sincerely welcome the reviewer comment. The manuscript is improved to read:

Main text, Page 20, line 399: The paragraph “The authors are aware that there exist many different sensor technologies to measure the human pulse wave, ranging from optic^{58,59} over ultrasonic⁶⁰ to force^{20,55,56,61–66} sensing approaches. The force-sensing approaches make use of piezo-resistive^{61,62}, piezo-electric^{55,56} or tribo-electric⁶³ effects, or of capacitive changes^{20,64–66}. Many of those devices are impressive with respect to their high sensitivity and ultrafast response time; for instance Yao et al. recently reported a piezo-resistive sensor with an impressive sensitivity $> 10^7 \Omega \cdot \text{kPa}^{-1}$ and a fast response time of 1.6 ms⁶¹. However, only a few can combine high sensitivity and fast response time with low power consumption, flexibility/conformability and biocompatibility.

The UFPT sensor technology excels for pulse wave monitoring in that it combines many aspects: it is self-powered (charge generation, not consumption), shows excellent mechanical stability (more than 1000 loading cycles), and has a high sensitivity ($> 10^3 \text{ pC N}^{-1}$) while offering ultrafast response ($\ll 20 \text{ ms N}^{-1}$). Furthermore, its ultraflexibility enables conformal attachment to various materials and surfaces as well as multilayer stacking even on 3D shaped carriers for further improvement in sensitivity ($> 10^4 \text{ pC N}^{-1}$).“ is added;

New References:

58. Rachim, V. P. & Chung, W.-Y. Multimodal Wrist Biosensor for Wearable Cuff-less Blood Pressure Monitoring System. *Sci. Rep.* **9**, 7947 (2019).
59. Yokota, T. *et al.* Ultraflexible organic photonic skin. *Sci. Adv.* **2**, e1501856 (2016).
60. Wang, C. *et al.* Monitoring of the central blood pressure waveform via a conformal ultrasonic device. *Nat. Biomed. Eng.* **2**, 687–695 (2018).
61. Yao, H. *et al.* Near-hysteresis-free soft tactile electronic skins for wearables and reliable machine learning. *Proc. Natl. Acad. Sci.* **117**, 25352–25359 (2020).
62. Nguyen, T.-V. *et al.* MEMS-Based Pulse Wave Sensor Utilizing a Piezoresistive Cantilever. *Sensors* **20**, 1052 (2020).
63. Xu, L. *et al.* Self-powered ultrasensitive pulse sensors for noninvasive multi-indicators cardiovascular monitoring. *Nano Energy* **81**, 105614 (2021).
64. Kaisti, M. *et al.* Clinical assessment of a non-invasive wearable MEMS pressure sensor array for monitoring of arterial pulse waveform, heart rate and detection of atrial fibrillation. *npj Digit. Med.* **2**, 39 (2019).
65. Zang, Y. *et al.* Flexible suspended gate organic thin-film transistors for ultra-sensitive pressure detection. *Nat. Commun.* **6**, 6269 (2015).
66. Schwartz, G. *et al.* Flexible polymer transistors with high pressure sensitivity for application in electronic skin and health monitoring. *Nat. Commun.* **4**, 1859 (2013).

Comment (#2-5): *In terms of the variance in the performance of the organic devices, it would be helpful to give a histogram of the rectifying performance across a few devices.*

Reply to comment (#2-5): Thank you for this comment. A histogram of the rectifying performance of the OTFT-based diodes for three different W/L ratio's are added to Fig. 13 (Fig. S13 c) and the corresponding mean values with standard deviation are added to Table S3. The text is modified according to:

Main text, Page 24, line 463: The sentence "This is illustrated in Fig. S13c, where the rectifying ratios of 31 OTFT-based diodes for three different W/L ratios (27 mm/12 μm , 7 mm/12 μm , 0.5 mm/12 μm) are plotted as a histogram. One can clearly see that the average rectifying ratio increases with increasing W/L." is added.

Supplementary Information, Page 16, lines 216-223: Table is modified.

Supplementary Table S3 | Comparison of the performance parameter of vertical diodes and OTFT-based diodes, both with DNTT as the active semiconducting layer.

	Gate dielectric	OSC	$J^{\text{a)}}$ (mA cm^{-2})	$V_{\text{T}}^{\text{b)}}$ (V)	Rectifying ratio ^{c)}	$V_{\text{break}}^{\text{d)}}$ (V)
Vertical diode with PFBT	-	DNTT	10^4 @ 5V	0.3–1.0	$>10^6$	>-15
OTFT-based diode (W=27mm)	AlO_x + SAM	DNTT	105 @ 2V	<0.1	$1.4 \cdot 10^7$ ($0.8 \cdot 10^7$)	>-5
OTFT-based diode (W=7mm)	AlO_x + SAM	DNTT	75 @ 2V	<0.1	$3.6 \cdot 10^6$ ($3 \cdot 10^6$)	>-5
OTFT-based diode (W=0.5mm)	AlO_x + SAM	DNTT	65 @ 2V	<0.1	$6.4 \cdot 10^5$ ($4.5 \cdot 10^5$)	>-5

^{a)} Current density range of vertical Schottky and OTFT-based diodes at forward voltages of 5 V and 2 V, respectively. The channel length of the OTFTs is 12 μm , and the channel width varies between 0.5 mm and 27 mm; the area of the vertical diodes is 0.025 mm^2 and for the OTFT-based diodes, it is between 0.018 and 0.65 mm^2 . ^{b)} V_{T} is the transition voltage; ^{c)} Rectifying ratio is defined as a ratio of the current in the 'on' ($V = 5$ V) and 'off' states ($V = -5$ V) for the vertical and ± 2 V for the OTFT-based diode. For the OTFT-based diodes with channel width of 0.5 mm, 7 mm and 27 mm the rectifying values are averaged over 15/10/6 devices, respectively, with the standard deviation values given in brackets; and ^{d)} V_{break} is the reverse breakdown voltage.

Supplementary Information, Page 17, lines 234-242: Figure S13c is added and the figure caption is improve to read:

Supplementary Figure S13 | OTFT-based organic diodes. (a) AC input voltage V_{in} (red) and DC output voltage V_{out} (black) after rectification with an OTFT-based **full wave rectifier** circuit (OFWR) and connection to a $1 \text{ M}\Omega$ resistor. The OTFTs in the OFWR circuit have an AlO_x + SAM gate dielectric, a DNTT semiconductor and a $W/L = 7 \text{ mm}/12 \mu\text{m}$. (b) Time dependence of the normalized output voltage $V_{dc, norm}$ of the OFWR for $V_{in} = 3 \text{ V} \sin(\omega \cdot t)$, $f = 0.1 \text{ Hz}$, and $C = 10 \mu\text{F}$. After 2 h and 45 min continuous operation, $V_{dc, norm}$ was reduced by not more than 5%. (c) Histogram of the rectifying ratios of 31 OTFT-based organic diodes with a channel length of $12 \mu\text{m}$ and varying channel widths – 15 diodes have a channel width of 0.5 mm (orange), ten diodes have a channel width of 7 mm (green), and six diodes have a channel width of 27 mm (purple).

List of further changes in the main text of the revised manuscript:

(1) Page 3:

Lines 63-65, the sentence “A more straightforward way to realize compliant sensors, nanogenerators and energy storage elements consists of integrating them on ultrathin substrates via spin coating or printing thus making them ultraflexible.” is changed to “A scalable method for easy realisation of compliant sensors, nanogenerators and energy storage elements is to integrate them on ultra-thin substrates by spin coating or printing, thus making them ultraflexible.”; **reason:** improving readability and linguistic expression

(2) Page 4:

Line 93, the word “they” is added;

(3) Page 5:

Line 110, the word “enabling” is changed to “which enables”;

(4) Page 7:

Line 122, the words “ultraflexible” and “UEHD” are added;

Line 135, the part of the sentence “ ... and induce a macroscopic polarization effect throughout the sample volume. ” is changed to “...in the entire sample volume and thereby induce a macroscopic polarization. “;

reason: improving readability and linguistic expression

Line 142, the words “values for the” are added;

(5) Page 8:

Line 158, the word “the” is added;

(6) Page 10:

Lines 205-206, “which is the mean value and standard deviation of ten transducers” is changed to “mean value and standard deviation calculated for ten transducers “;

Line 207, the words “ $120\text{ }^{\circ}\text{C} \leq T_A \leq 80\text{ }^{\circ}\text{C}$ ” is changed to “ $80\text{ }^{\circ}\text{C} \leq T_A \leq 120\text{ }^{\circ}\text{C}$ “;

Line 214, the words “the” is changed to “a “;

(7) Page 11:

Lines 222-225, the sentence “By comparing Fig. 2c and 2d, it became obvious that X_c – and consequently the number and size of crystallites – increased drastically above 70°C; a ferroelectric behaviour was observed at precisely this same annealing temperature.” is changed to “, where X_c – and consequently the number and size of crystallites – is drastically increased and ferroelectric behaviour is observed.”; **reason:** improving comprehensibility and readability

Line 228, the words “ $\epsilon_{r,poled} < \epsilon_r$ ” is changed to “ $\epsilon_{r, after\ poling} < \epsilon_{r, before\ poling}$ ”;

Line 230, the words “, defined as $\Delta\epsilon_r = \epsilon_{r, after\ poling} - \epsilon_{r, before\ poling}$,” are added;

(8) pages 11/12:

Lines 243-244, the part of the sentence “and $d\sigma_{11}$ or $d\sigma_{22}$, respectively, (here, tensile stress longitudinal to the film plane) induce changes in the...” is changed to “and $d\sigma_{11}$ or $d\sigma_{22}$ (here, tensile stress longitudinal to the film plane), respectively, induce changes in the...”;

(9) Page 15:

Line 291, the word “enhanced” is changed to “enhance”;

Line 304, “to surpass” is changed to “surpasses”; **reason:** Correcting a grammatical error

(10) Page 16:

Line 329, the word “revealed” is changed to “shows”;

reason: Correcting a grammatical error and improving linguistic expression

(11) Page 17:

Lines 338-340, the word “is” is changed to “was”; **reason:** Tense correction

Lines 348-349, “during repetitive tensile/compressive and pressure/release strain loadings, respectively.” is changed to “during repetitive longitudinal (strain/release) and transversal (push/release) loadings”; **reason:** Improving comprehensibility and linguistic expression

Line 350, the words “strain” and “load” are deleted;

(12) Page 18:

Line 373, “pulse wave signal” is changed to “recorded signal”;

reason: Improving linguistic expression

(13) Page 19:

Lines 382-384, the sentences "... (here, $AI \sim 56\%$) are derived and (b) the blood pressure of the human arteria in the neck by measuring the signal delay Δt . The pulse wave velocity PWV can be derived from the signal delay for a given sensor distance Δx ." are changed to "... (here: $AI \sim 56\%$) can be measured as well as (b) the blood pressure of the human arteria in the neck via the pulse wave velocity PWV . PWV can be determined by measuring the signal delay Δt for a given sensor distance Δx .";

reason: Improving comprehensibility and linguistic expression

Line 389, "can" is changed to "could";

(14) Page 21:

Line 403, the word "UEHD" is changed to "UEHDs"; **reason:** Improving grammar

Line 410, the word "S11" is changed to "S12";

Line 416, the word "Chapter 11" is added;

Line 419, the word "a via" is changed to "vias"; **reason:** Improving comprehensibility

(15) Page 22:

Lines 434-442, the sentences "The black curve of the diode $I(V)$ characteristics in the right plot corresponded to the black OTFT transfer curve in the left plot. In addition, a red and a blue curve of the diode $I(V)$ characteristics are shown indicating a strong relation to the measured onset voltage of the transistor transfer characteristics. A highly negative onset voltage yielded a positive transition voltage in the diode (blue curve in Fig. 6c) resulting in a parasitic voltage drop across the diode. That would decrease its rectifying performance. In contrast, a transistor onset that is overly positive increased the diode's off-current (red curves in Fig. 6c), which again strongly reduced its rectification ratio." are changed to "The black curve of the diode $I(V)$ characteristics (Fig. 6c, right plot) corresponds to the black curve of the OTFT transfer characteristics (Fig. 6c, left plot). The measured onset voltage of the transistor transfer characteristics has a strong relation to the diode performance as illustrated by two other diode $I(V)$ curves plotted in red and blue. A highly negative transistor onset voltage would yield a positive transition voltage in the diode (blue curve in Fig. 6c right) resulting in a parasitic voltage drop across the diode. That would decrease its rectifying performance. In contrast, a transistor onset that is overly positive (transistor transfer curve in red in Fig. 6c left) increased the diode's off-current (red curve in Fig. 6c right), which again strongly reduced its rectification ratio.";

reason: Improving grammar,

comprehensibility and linguistic expression.

(16) Page 24:

Line 463, the sentence “FWR” is changed to “(OFWR)”;

Line 463, the words “made of” are added;

reason: Improving linguistic expression

Line 466 and 470, the word “S12” is changed to “S13”;

Line 468, the word “DC” is added;

Line 477, the word “S13” is changed to “S14”;

(17) Pages 24/25:

Line 485 and 488, the sentence “According to the FEM simulations (see Fig. S7) for a given bending excitation an ultraflexible ferroelectric film is subjected to considerably stronger mechanical deformation and higher longitudinal strain forces, which in turn will significantly increase the output signal from the transducers.” is deleted;

(18) Page 25:

Lines 488-492, the sentence “We tested three different bending modes: (A) continuous bending by hand, the results are shown in Fig. 7b and Fig. S14; (B) controlled continuous bending over a rail with results shown in Fig. 7 and Fig. S15, and (C) continuous pressing with the fingertip on the bent transducer Fig. 7b,d and Fig. S16.” is modified to “We tested three different bending modes: (A) continuous bending by hand with results shown in Fig. 7b and Fig. S15; (B) controlled continuous bending over a rail, see Fig. 7 and Fig. S16, and (C) continuous pressing with the fingertip on the bent transducer, see Fig. 7b,d and Fig. S17.”;

reason: Improving comprehensibility

Line 505, “S14” is changed to “S15”;

(19) Page 26:

Line 516, “release” is changed to “releasing motion”; **reason:** Improving readability

Line 521, “300” is changed to “450”;

Line 527, “S17” is changed to “S18”;

(20) Page 27:

Line 536, “Finally, we could...” is changed to “We further demonstrated...”;

Line 538 & Line 539, “S16” is changed to “S17”;

(21) Page 28:

Lines 545-549, “The output power density P_{out} are plotted as a function of load resistance R_L and compared to those from mode A and C excitation. (c) In the right-hand graph, the charging curve of a capacitor is plotted; with the UFPT being connected to the ultrathin OFWR, which then charges the capacitor ($C = 10 \mu\text{F}$).” is changed to “The output power density P_{out} of mode B is plotted as a function of load resistance R_L (black curve) and compared to those from mode A (red curve) and mode C (blue curve). (c) Charging curve of a capacitor for an UFPT excited in mode B. The UFPT is connected to the ultrathin OFWR, which then charges the capacitor ($C = 10 \mu\text{F}$).”;

reason: Improving clarity

Line 551, “levels” is added;

(22) Page 29:

Line 576, “and boosts the energy conversion efficiency” is added;

Line 576, “easily” is deleted;

Line 577, “in a straightforward way” is added;

Lines 577-579, “Owing to their ultraflexible and good adhesion properties, they can easily be stacked in multiple layers, thereby multiplying their respective sensing/harvesting performance proportionally.” Is changed to “Owing to their ultraflexible and good adhesion properties, they can be easily stacked, thereby multiplying the respective sensing/harvesting performance proportionally.”;

Line 587, “energy” is added;

(23) Page 30:

Lines 599-602, “Owing to its superior integration possibilities, the presented ultraflexible energy harvesting and sensing technology paves the way to many potential ‘energy self-sufficient’ applications, ranging from robotics over artificial e-skin and wearable electronics to remote sensor networks.” is deleted

(24) Page 32:

Line 634, “(Strictly speaking, the monitored surface charge is related to the dielectric displacement D rather to the polarization P . However, since the displacement during poling is dominated by the spontaneous polarization stemming from the microscopic dipoles, we follow

here the tradition of labelling the ordinate of the hysteresis plots with P rather than D .)” is added.

(25) Page 33:

Line 680, “S15” is changed to “S16”;

(26) Page 35:

Line 718, “As to the FEM simulation of the transducer response and energy conversion efficiency in the bending mode B, a detailed description is given in the Supplementary Information, Chapter 19.” is added;

(27) Page 36:

Line 740, “S17” is changed to “S18”;

Line 757, “S13” is changed to “S14”;

List of further changes in the Supplementary Information of the revised manuscript:

(28) Page 2:

Line 13, in Fig. S1 “remanent” is changed to “remnant” and “coercitive” to “coercive”;

(29) Page 4:

Line 47, the words “at 1 kHz” are added;

(30) Page 6:

Line 68, In Fig. S4d “ $-\Delta\epsilon$ ” is changed to “ $\Delta\epsilon$ ” and “-” is added to the values on this axis;

Line 73: “ ϵ_r after the” is changed to “ $\Delta\epsilon_r = \epsilon_{r, \text{before poling}} - \epsilon_{r, \text{after poling}}$ due to”

(31) Page 7:

Line 85, the word “apparent” are added;

(32) Page 14/15:

Lines: 173, 185, 189, 193, 197, 205, the word “S11” is changed to “S12”;

(33) Page 18:

Line 247, “12” is changed to “13”; Line 251, “S13” is changed to “S14”;

(34) Page 19:

Line 257, “13” is changed to “14”; Line 262, “S14” is changed to “S15”;

(35) Page 20:

Line 271, “14” is changed to “15”; Line 274, “S15” is changed to “S16”;

(36) Page 21:

Line 282, “15” is changed to “16”; Line 286, “S16” is changed to “S17”;

(37) Page 22:

Line 298, “16” is changed to “17”; Line 302, “S17” is changed to “S18”;

Having addressed all comments, we fervently hope that the revised manuscript now qualifies for publication in Nature Communications.

Reviewer #1 (Remarks to the Author):

The authors have addressed my comments appropriately and I now recommend publication of the paper in its present form.

Reviewer #2 (Remarks to the Author):

The authors have adequately addressed my concerns and made improvements to the paper.